# Evolution of complexity in the zebrafish synapse proteome

Àlex Bayés[1,2], Mark O. Collins[3], Rita Reig-Viader[1,2], Gemma Gou[1,2], David Goulding[4], Abril Izquierdo[5], Jyoti S. Choudhary[6], Richard D. Emes[5,7] & Seth G.N. Grant[8]

The proteome of human brain synapses is highly complex and is mutated in over 130 diseases. This complexity arose from two whole-genome duplications early in the vertebrate lineage. Zebrafish are used in modelling human diseases; however, its synapse proteome is uncharacterized, and whether the teleost-specific genome duplication (TSGD) influenced complexity is unknown. We report the characterization of the proteomes and ultrastructure of central synapses in zebrafish and analyse the importance of the TSGD. While the TSGD increases overall synapse proteome complexity, the postsynaptic density (PSD) proteome of zebrafish has lower complexity than mammals. A highly conserved set of ∼1,000 proteins is shared across vertebrates. PSD ultrastructural features are also conserved. Lineage-specific proteome differences indicate that vertebrate species evolved distinct synapse types and functions. The data sets are a resource for a wide range of studies and have important implications for the use of zebrafish in modelling human synaptic diseases.

[1] Molecular Physiology of the Synapse Laboratory, Biomedical Research Institute Sant Pau (IIB Sant Pau), Sant Antoni Maria Claret 167, 08025 Barcelona, Spain. [2] Universitat Autònoma de Barcelona, Cerdanyola del Vallès, 08193 Bellaterra, Spain. [3] Department of Biomedical Science, The Centre for Membrane Interactions and Dynamics, University of Sheffield, Western Bank, Sheffield S10 2TN, UK. [4] Pathogen Genomics, Wellcome Trust Sanger Institute, Hinxton CB10 1SA, UK. [5] School of Veterinary Medicine and Science, University of Nottingham. Sutton Bonington Campus, Leicestershire LE12 5RD, UK. [6] Proteomic Mass Spectrometry, The Wellcome Trust Sanger Institute, Hinxton, Cambridgeshire CB10 1SA, UK. [7] Advanced Data Analysis Centre, University of Nottingham, Sutton Bonington Campus, Leicestershire LE12 5RD, UK. [8] Genes to Cognition Programme, Centre for Clinical Brain Science, University of Edinburgh, Edinburgh EH16 4SB, UK. Correspondence and requests for materials should be addressed to À.B. (email: abayesp@santpau.cat) or to S.G.N.G. (email: Seth.Grant@ed.ac.uk).

Synapses are the hallmark of the central nervous system. Although synapses were originally considered to be simple connectors between neurons, they are now recognized to be highly sophisticated computational units built from proteomes containing in excess of 1,000 proteins that regulate the behavioural repertoire[1–4]. Proteomic studies revealed that genetic disruption of postsynaptic density (PSD) proteins results in over 130 human mental and neurological disorders[5,6]. These disorders are now known as synaptopathies[7,8] and include complex genetic disorders such as intellectual disability, autism spectrum disorders and schizophrenia[5,9,10].

Comparative proteomic and genomic approaches have been used to study the evolutionary origins of vertebrate postsynaptic complexity. The major classes of vertebrate synapse proteins evolved in unicellular eukaryotes, and these proteins were recruited into synapses in early invertebrates[11]. Subsequent whole-genome duplications (WGD) have played a major role in generating and shaping the complexity of vertebrate synapse proteomes. Two WGDs in early vertebrates resulted in a major expansion of the synapse proteome, which distinguishes them from invertebrate synapses. This event, which occurred ∼550 million years ago[12], generated ohnologues (paralogues arising from WGDs) that subsequently diversified, potentially contributing to the enhanced cognitive abilities and behavioural repertoire of vertebrates[2]. Importantly, there was another WGD ∼300 million years ago[13] in the major clade of the fish lineage known as the teleost-specific genome duplication (TSGD). Although the TSGD increased the number of protein-coding genes in zebrafish[14] and other fish compared to mammals, it is unknown whether this influenced synapse proteome complexity. While the mammalian synaptic proteome has been characterized, the absence of similar data from teleosts limits our knowledge on the evolution of synapses in vertebrates and the roles of WGDs.

The freshwater fish *Danio rerio* (zebrafish) is a teleost that is now widely used in neuroscience for modelling human genetic brain disorders including those that disrupt synapse proteins[15–17]. Here we report the first characterization of the ultrastructure, proteome composition and evolution of zebrafish central nervous synapses. The proteome of synaptosomes and the PSD of zebrafish and mice were analysed in parallel and their complexity compared. Surprisingly, despite zebrafish having an extra WGD the PSD proteome was less complex when compared with mammals. We identify a core 'vertebrate PSD' (vPSD) that corresponds to the ancestral postsynaptic machinery common to all vertebrates and conserved ultrastructural features. We have also identified proteins that are only present in the mouse proteome, representing molecular innovations either acquired by mammals after divergence of the fish lineage or specifically lost from the fish lineage. We have made these data freely available in a database and web resource that includes links to a wide variety of related biomedical data sets (http://www.genes2cognition.org/publications/zebrafish-prot/).

## Results

### Ultrastructure of zebrafish synapses.
Before analysing the synapse proteome we performed an ultrastructural analysis of central synapses in zebrafish to address two questions: do zebrafish synapses contain PSDs (a prerequisite for their biochemical isolation), and, if so, do they show any morphological features that are conserved with mammals? Moreover, to our knowledge, the ultrastructure of zebrafish brain synapses has not been previously described, although studies in other bony fish species were reported several decades ago[18–22]. We therefore examined the four main regions of the zebrafish brain with transmission electron microscopy (Fig. 1, Supplementary Note 1,

Supplementary Figs 1–5 and Supplementary Table 1). In olfactory bulb, telencephalon, midbrain and hindbrain (Fig. 1b) asymmetric synapses presented structures equivalent to mammalian PSDs (Fig. 1c). While PSDs were identified across the entire transverse sections of olfactory bulb and telencephalon, in midbrain and hindbrain these were restricted to the optic tectum and cerebellar corpus, respectively, which are layered, cortex-like structures located in the most dorsal part of the brain (Supplementary Figs 3 and 4). Asymmetric synapses from the olfactory bulb were morphologically similar to those found in mammals and other bony fish species[20,21] (Fig. 1c,d and Supplementary Fig. 1). The zebrafish olfactory bulb presented the characteristic dendrodendritic synapses[23] of this brain region, which contains synaptic vesicles on both sides of the synaptic cleft (Fig. 1d and Supplementary Fig. 1). Synapses in the telencephalon also showed the prototypical characteristics of mammalian synapses with PSDs present in spine-like structures (Fig. 1e and Supplementary Fig. 2). At the level of the optic tectum, PSDs were mainly present in the medial layers of this structure (Supplementary Fig. 3b, orange delimited area). Presynaptic boutons in the optic tectum appeared to make synaptic contacts directly on dendritic shafts rather than on spines as suggested by the small diameter of postsynaptic elements and the presence of microtubule-like structures beneath the PSD (Fig. 1f and Supplementary Fig. 3).

Synapses in the cerebellar corpus showed distinct features and could be classified into different types. Although we observed synapses with flat PSDs and aligned pre- and postsynaptic membranes (Fig. 1g and Supplementary Fig. 5a), these were a minority (13%, Supplementary Fig. 5c). Most synapses presented highly curved PSDs and a presynaptic element surrounding the postsynaptic spine (Fig. 1h–i and Supplementary Figs 4 and 5b). In mammals, cerebellar glutamatergic synapses present a presynaptic element that partially surrounds the postsynaptic structure[24]; however, this feature is greatly enhanced in the zebrafish cerebellum. When measuring the arch length of curved PSDs and their postsynaptic elements these can be divided into short and long PSDs (Supplementary Fig. 5d–i). Thus, at the zebrafish cerebellar corpus we could identify three PSD shapes: 'flat', 'round-short' and 'round-long' (Supplementary Fig. 5 and Supplementary Table 1). Finally, we compared PSD length and the area between all four brain regions and found that flat PSDs and cerebellum round-short PSDs have similar sizes and areas (Supplementary Fig. 5j–l), while telencephalon and cerebellum round-long PSDs were larger, with the cerebellum PSDs being the largest. These studies show a diversity of synapse ultrastructure in zebrafish and characteristic features shared with mammalian synapses.

### Proteomic profiling of zebrafish and mouse synapses.
We purified synaptosomal (SYN) and PSD fractions in triplicate from mouse and zebrafish brains using identical protocols, generating equivalent yields and expected PSD enrichments (Supplementary Fig. 6). These data indicate that the performance of the biochemical methods was equivalent between species, although stochastic effects on small numbers of proteins cannot be fully ruled out. Quantitative mass spectrometry-based proteomic analysis (see Methods, Supplementary Note 2 and Supplementary Fig. 7) identified a total of 3,579 and 3,840 proteins in triplicate for mouse and zebrafish, respectively (Fig. 2a and Supplementary Data 1 and 2). In order to define which proteins are enriched/depleted in PSDs compared to their parent SYN fractions, we used a label-free quantification and statistical analysis of the mass spectrometry data. This identified proteins significantly enriched in SYN (depleted from PSD), which were removed from the final

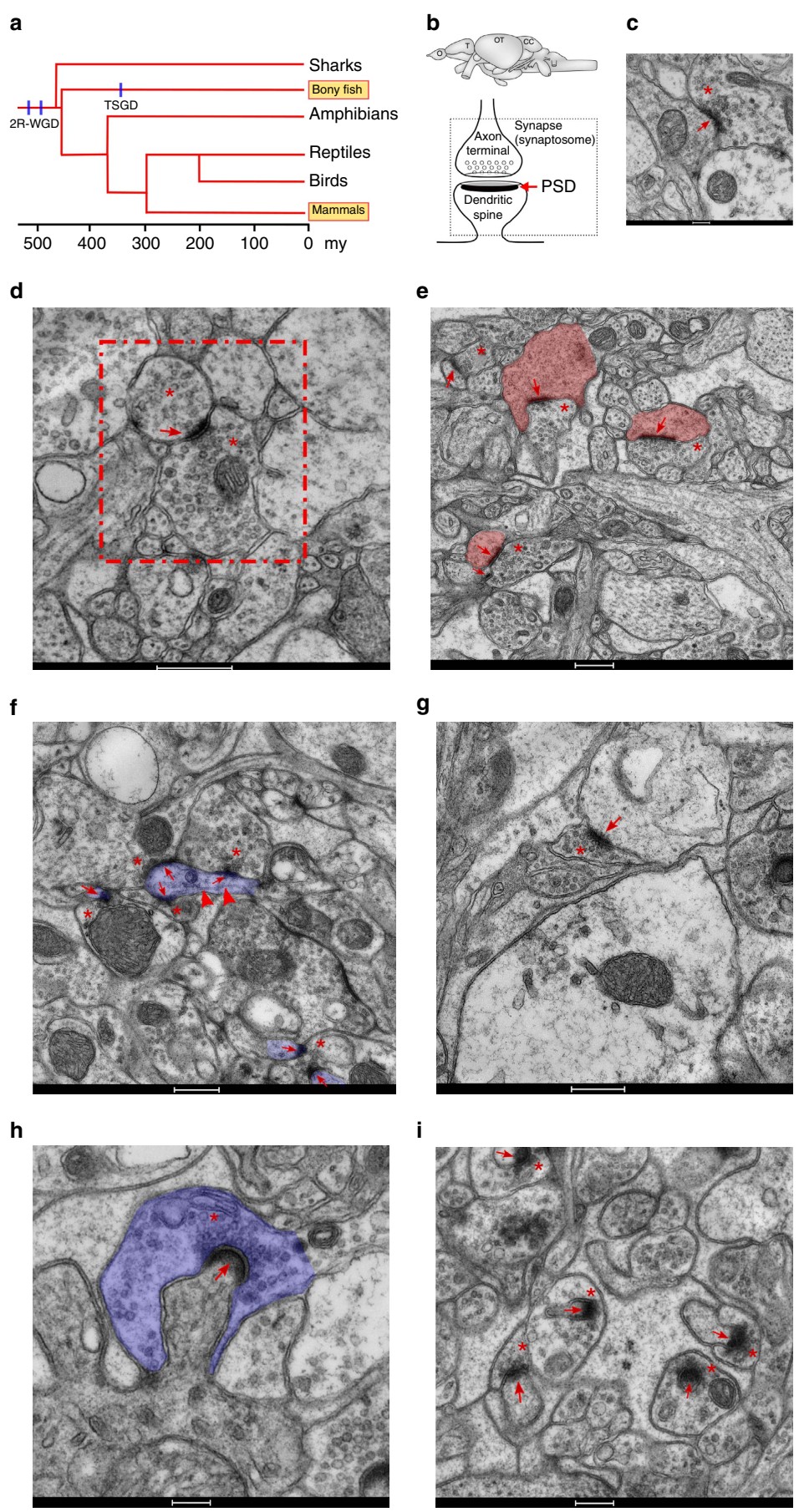

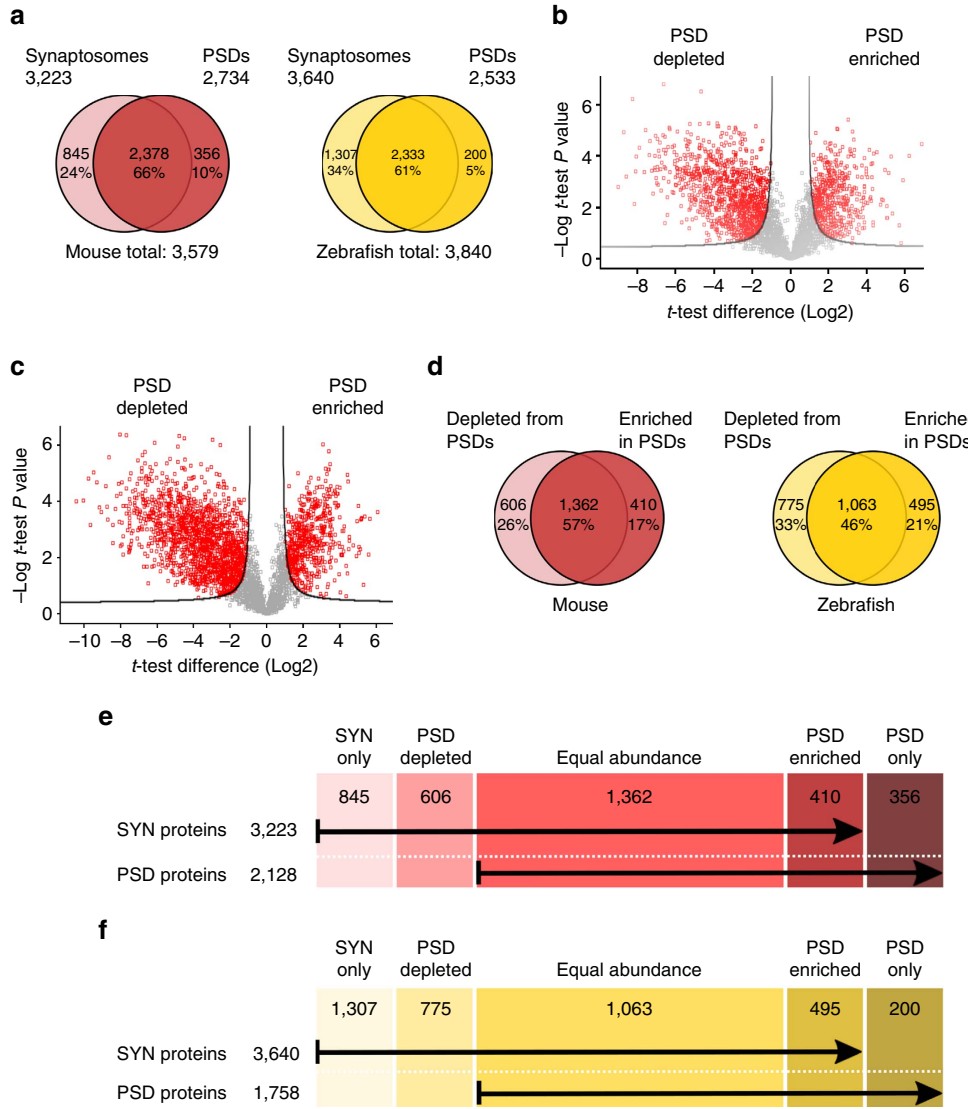

**Figure 2 | Mouse and zebrafish synaptic proteome.** (**a**) Venn diagrams of mouse (red) and zebrafish (yellow) proteins identified in synaptosomal and PSD preparations, indicating the percentage of proteins found in both fractions. The total number of proteins identified in each species is also indicated. (**b,c**) Volcano plots showing quantitative enrichment and depletion of proteins between synaptosomes and postsynaptic densities purified from mouse (**b**) and zebrafish brain (**c**). Enriched or depleted proteins were identified from statistical analysis of triplicate PSD and synaptosome data sets for each species using *t*-testing and a Permutation-based false discovery rate of 0.05. (**d**) Venn diagrams of mouse (red) and zebrafish (yellow) proteins found depleted or enriched in the PSD when compared with the synaptosomal fraction. The percentage of proteins found with equal abundance at the synaptosomal and PSD fraction is also indicated. (**e**) Scheme indicating the number of mouse synaptosomal and PSD proteins found only in one of the two fractions, depleted or enriched at the PSD or found in equal abundance in both fractions. (**f**) Scheme indicating the number of zebrafish synaptosomal and PSD proteins found only in one of the two fractions, depleted or enriched at the PSD or found in equal abundance in both fractions.

**Figure 1 | Transmission electron microscopy of zebrafish asymmetric synapses in four different brain areas.** (**a**) Evolutionary tree of the vertebrate lineage with timescale in million years (my). The occurrence of the two WGD events common to all vertebrates (2R-WGD) and specific to teleosts (TSGD) are indicated by blue lines. (**b**) Schematic representation of the zebrafish brain with the four regions studied (CC, cerebellar corpus; O, olfactory bulb; OT, optic tectum; T, telencephalon) and of an excitatory synapse. Synaptosomes are formed by the axon terminal and the dendritic spine, which are separated from their corresponding neurons during tissue processing. The location of the PSD is also indicated. (**c**) Asymmetric synapse from the olfactory bulb. A red asterisk and a red arrow indicate the location of presynaptic vesicles and the PSD, respectively. Scale bar, 200 nm. (**d**) An asymmetric dendrodendritic synapse of the olfactory bulb (framed by a red dotted square) is shown. Asterisks indicate pre- and post-synaptic vesicles. The PSD is indicated by a red arrow. Scale bar, 500 nm. (**e**) Asymmetric synapses from the telencephalon. Red asterisks and arrows indicate the location of presynaptic vesicles and PSDs, respectively. The area corresponding to postsynaptic spine-like structures is filled with pink. Scale bar, 500 nm. (**f**) Asymmetric synapses from the optic tectum. Red asterisks and arrows indicate the location of presynaptic vesicles and the PSD, respectively. Red arrowheads indicate microtubule location within thin dendritic-like projection. The area of a thin dendritic-like projection, where synapses are formed, is filled with purple. Scale bar, 500 nm. (**g**) Flat (standard) asymmetric synapse from the cerebellar corpus. A red asterisk and a red arrow indicate the location of presynaptic vesicles and the PSD, respectively. Scale bar, 500 nm. (**h**) Asymmetric synapse from the medial part of the cerebellar corpus showing the extent at which the presynaptic element (highlighted in purple) surrounds the dendritic spine. Scale bar, 200 nm. (**i**) Micrograph displaying the morphology of most abundant asymmetric synapses from the cerebellar corpus. Red asterisks and arrows indicate the location of presynaptic vesicles and the PSD, respectively. Scale bar, 500 nm.

list of PSD components as potential purification contaminants (Fig. 2b–d and Supplementary Data 1 and 2). Thus, we document 3,223 and 2,128 proteins in mouse SYN and PSD fractions, respectively, and 3,640 and 1,758 proteins in the corresponding zebrafish structures (Fig. 2e,f). The greater number of zebrafish SYN proteins compared to mouse likely reflects the greater number of protein-coding genes in the zebrafish genome, which is supported by the finding that the proportion of SYN genes relative to genome size is 14% in both species. Surprisingly, the zebrafish PSD proteome was 17% smaller than mouse ($P < 1E - 06$, binomial test). This difference is still significant if the PSD is defined as the sum of zebrafish proteins exclusively found in the PSD or significantly enriched in it ($P = 0.0003$, binomial test). The zebrafish PSD was only 48% of the SYN proteome compared to the 66% in mouse. Hence, despite the TSGD and the concomitant expansion of the zebrafish synapse proteome, the zebrafish PSD is of smaller size than that found in mammals.

**Teleost genome duplication expanded synapse protein families**. Many well-known families of synaptic proteins were found with an expanded number of ohnologues in the zebrafish. For example, zebrafish show twice as many ionotropic glutamate receptor subunits in the NMDA (N-methyl-D-aspartate) and AMPA (α-amino-3-hydroxy-5-methyl-4-isoxazole propionic acid) families and more scaffold proteins in the PSD95/Dlg family (six in zebrafish and four in mice, Supplementary Data 2). We therefore asked whether family expansion was a common feature among SYN and PSD proteins in zebrafish. Using the Ensembl Families classification we found that both zebrafish SYN (Fig. 3a) and PSD (Fig. 3b) proteomes contain protein families with a significantly higher number of components.

We next asked whether synapse genes were more likely to be retained after the TSGD than genes expressed elsewhere in the brain or in other tissues. We calculated the fraction of proteins belonging to the orthology types (zebrafish:mouse: 1:1, 1:many, many:1, many:many and unique to each species; Fig. 3c). The many:1 category is increased in synaptosomes and PSD protein families compared to the genome-wide ratio, indicating that synapse genes have been retained at higher frequencies after the TSGD than seen in the genome as a whole. To quantify these differences we determined the number of orthologues in each species and calculated the ratio of zebrafish:mouse orthologues (Fig. 3d). While most genes have a 1:1 ratio between species, a clear peak appears at 2:1 representing genes with double the number of orthologues in zebrafish compared to a small peak at 1:2 representing genes duplicated in mouse. This ratio is also seen in other teleosts but not in the Spotted Gar (Lepisosteus oculatus), a fish whose lineage diverged before the teleost-specific WGD[25] (Supplementary Fig. 8). Examples of the increased ratio of orthologues in key synaptic proteins among fish species but not in the Gar, and other vertebrate and invertebrate species, are shown in Fig. 3e. These data support the assumption that the trend for gene family expansion in zebrafish is a legacy of the TSGD rather than by specific loss of genes in mammalian genomes. The distribution of orthologue ratios for SYN and PSD proteomes is statistically different to the whole-genome proteomes, and the PSD is even statistically different to the brain proteome. No statistical difference was seen between SYN and PSD proteomes (Fig. 3d). These results show that following the TSGD zebrafish synapse proteome-encoding genes, especially PSD ones, were more frequently retained as duplicates.

**Functional complexity of the synapse proteome**. To understand the functional implications of the different complexity of zebrafish and mouse synapse proteomes, we examined protein diversity. We first considered high-level categories corresponding to protein cellular location and molecular function from the ingenuity pathway analysis (IPA) knowledgebase functional classification system: no significant differences were observed between the percentages of protein location or function, indicating a general conservation of the molecular characteristics of SYN and PSD proteomes between species (Fig. 4a,b and Supplementary Data 3). To corroborate this finding we compared enriched functional categories from two other classification systems: the Gene Ontology (GO-Slim) and the Panther Protein Class ontology. Most of the significantly enriched functional categories were found in both species (Fig. 4c and Supplementary Data 3), supporting the conclusion that the overall functionality of mammalian and teleost synapse proteomes is conserved.

Nevertheless, as we observed differences in the number of PSD proteins between species, we asked whether there might be differences in the number of protein families, using the Ensembl Protein Family annotation. Zebrafish showed fewer families in both SYN and PSD proteomes even when accounting for proteome size (Table 1 and Supplementary Data 4). To correct for possible genome annotation differences between species, we obtained Ensembl Protein Family IDs from mouse orthologues of zebrafish proteins and repeated the analysis, obtaining the same results (Table 1). The number of zebrafish PSD families was significantly lower than expected for zebrafish PSD proteins or mouse orthologues of zebrafish PSD proteins (Table 1). Thus, the lower PSD complexity in zebrafish results from fewer protein families.

Since protein domains contribute to functional complexity, we examined domain composition (number of unique protein domain types/protein, Supplementary Data 5) in synaptic proteins, mammalian and zebrafish synaptic brain proteomes[26–29] and all mouse and zebrafish coding proteins (Supplementary Data 6). We did not find a statistically significant difference of domain complexity between species for any of the proteomes. However, within species we found that the SYN and PSD proteomes do have higher complexity than brain or genome (number of unique protein domain types/protein mouse: genome = 1.46, brain = 1.50, SYN = 1.66, PSD = 1.64; zebrafish: genome = 1.46, brain = 1.57, SYN = 1.67, PSD = 1.75), suggesting that SYN and PSD represent specialized proteomes with higher functional complexity compared to the brain or whole proteome. Cumulative distributions (Fig. 4d,e) show significant increase in unique domains per protein in SYN and PSD compared to brain and genome data sets.

**Species specialization in the PSD**. We next focussed our attention on identifying those biological functions that were specific to either zebrafish or mouse PSDs. The zebrafish-specific PSD (Zf-sPSD, 523 proteins) and mouse-specific PSD (Mm-sPSD, 745 proteins) proteomes were examined for enrichment of GO terms from biological process and Cellular Component categories. To avoid potentially misleading differences between species arising from the biochemical fractionation, for a protein to be considered species-specific it had to be absent from both the PSD and SYN proteomes in the reciprocal species. To account for the possibly less complete annotation of the zebrafish genome, the enrichment analysis with zebrafish proteins was done twice, first using Zf-sPSD proteins against the zebrafish genome and later using mouse orthologues of Zf-sPSD against the mouse genome. The final list of zebrafish-enriched terms corresponded to the sum of terms enriched in both analyses.

A large difference in the number of significantly enriched terms was found between the two species-specific proteomes: the Mm-sPSD presented 97 biological process and 66 cellular component-

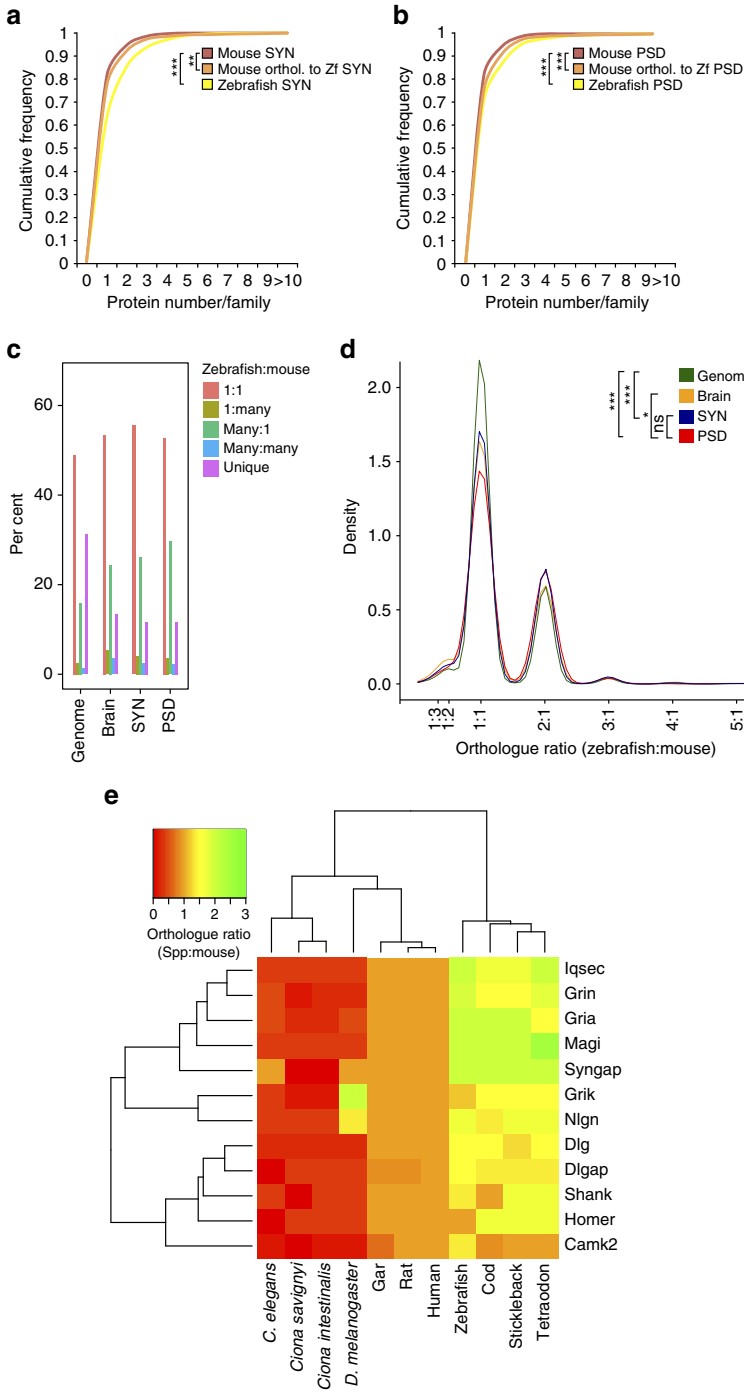

**Figure 3 | Expansion and retention of synaptic proteins after teleost-specific whole-genome duplication.** (**a**) Cumulative frequency plot of proteins found per Ensemble Family among mouse, zebrafish and mouse orthologues of zebrafish SYN proteins. Kruskal–Wallis test was used to calculate significance between distributions (**$P < 0.01$ and ***$P < 0.0001$). (**b**) Cumulative frequency plot of proteins found per Ensembl Family among mouse, zebrafish and mouse orthologues of zebrafish PSD proteins. Kruskal–Wallis test was used to calculate significance between distributions (**$P < 0.01$ and ***$P < 0.0001$). (**c**) Distribution of orthology types between zebrafish and mouse. For each protein in the proteome, the corresponding gene was identified and orthologous genes between species were determined using the biomaRt bioconductor package[67]. For each gene the zebrafish:mouse ratio of orthologues was determined and characterized as 1:1, 1:many, many:1, many:many or was unique to mouse or zebrafish. (**d**) Density of proteome by orthologue ratio (zebrafish:mouse). Statistically significant comparisons between pairs of distributions (two-tailed Kolmogorov–Smirnov test applied to distributions) are shown. ***$P < 0.001$ and *$P < 0.05$; nonsignificant comparisons are not shown. (**e**) A heatmap representation of homologue ratio between mouse and each species for key synaptic gene families. For each gene family and species the homologues of mouse genes were identified using the biomaRt bioconductor package[67]. Colours represent the relative expansion (yellow–green) or reduction (red) of gene families compared to the size seen in mouse. Orange represents a 1:1 ratio where family size is equal in mouse and other species. Dendrograms show the similarity of species and gene families.

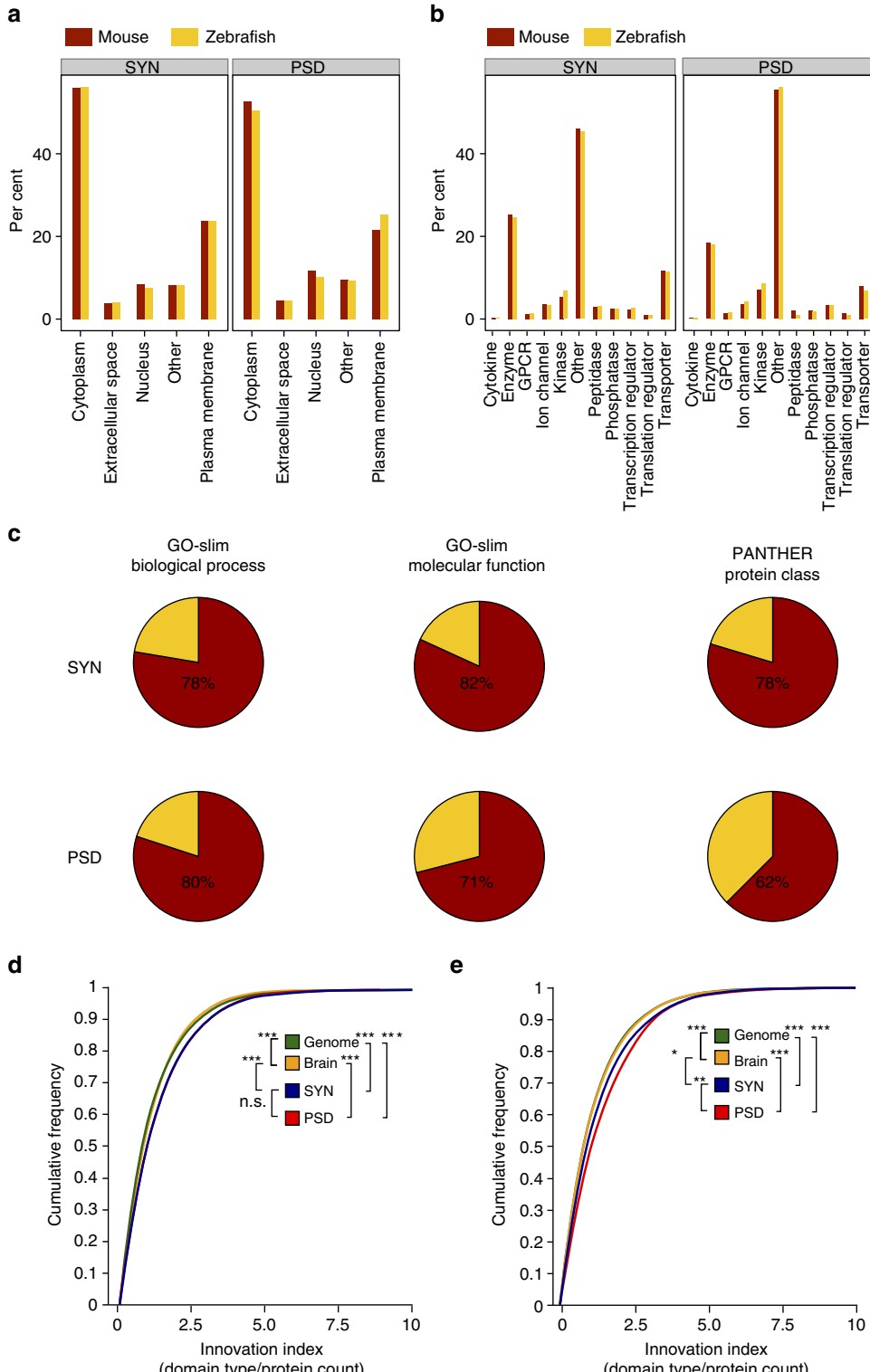

**Figure 4 | Functional similarity between mouse and zebrafish synaptic proteomes.** (**a**) Fraction of mouse (red) and zebrafish (yellow) synaptic proteins annotated to IPA cell location categories. (**b**) Fraction of mouse (red) and zebrafish (yellow) synaptic proteins annotated to IPA molecular function categories. (**c**) Overlap between Biological Process and Molecular Function GO-slim terms[68] and PANTHER Protein Classes[69] enriched in mouse and zebrafish. Top row shows data for synaptosomes and bottom row for PSD proteomes. The exact percentage of overlap is indicated. (**d**) Cumulative frequency distribution plots of individual protein innovation index (number of unique domain types per protein) for each proteome in mouse. Statistically significant comparisons between pairs of distributions (two-tailed Kolmogorov–Smirnov test applied to distributions) are shown at the legend. ***$P < 0.001$, **$P < 0.01$, *$P < 0.05$; nonsignificant comparisons are not shown. (**e**) Cumulative frequency distribution plots of individual protein innovation index (number of unique domain types per protein) for each proteome in zebrafish. Statistically significant comparisons between pairs of distributions (two-tailed Kolmogorov–Smirnov test applied to distributions) are shown at the legend. ***$P < 0.001$, **$P < 0.01$, *$P < 0.05$; nonsignificant comparisons are not shown.

**Table 1 | Protein families among mouse and zebrafish SYN and PSD proteins.**

|  | Proteins | Ensembl families | Families/protein |
|---|---|---|---|
| Mouse SYN | 3,223 | 2,552 | 0.79 |
| Zebrafish SYN | 3,640 | 2,354 | 0.65 |
| Mouse orthologues of zebrafish SYN | 3,138 | 2,410 | 0.77 |
| Mouse PSD | 2,128 | 1,688 | 0.79 |
| Zebrafish PSD | 1,758 | 1,089* | 0.62 |
| Mouse orthologues of zebrafish PSD | 1,556 | 1,128*** | 0.72 |

PSD, postsynaptic density; SYN, synaptosomal.
*$P < 0.05$ and ***$P < 0.0001$.

enriched terms, while only 17 and 8 were found in Zf-sPSD. Most (80%) Mm-sPSD proteins presented an orthologue in the zebrafish genome (Supplementary Note 3 and Supplementary Fig. 9), indicating that gene loss in zebrafish or gene gain in mouse is not the only factor driving PSD functional differences observed between species. Terms enriched in zebrafish were not obviously relevant to synaptic biology, whereas Mm-sPSD proteins were enriched in terms such as 'postsynaptic density', 'synapse' or 'regulation of synapse structure or activity' (Supplementary Data 7). This is consistent with specialized synaptic proteins found in mammalian synapses being absent from the Zf-sPSD. In addition to these proteins, most of the other mouse-enriched terms fall into a few functions embracing endocytosis, vesicle-mediated intracellular trafficking, protein localization to the plasma membrane and actin filament-based processes (Supplementary Data 7).

Among proteins involved in vesicle traffic and endocytosis, particularly noticeable was the differential presence at the mouse PSD of many proteins forming SNARE complexes. These included syntaxins, synaptobrevins (vamps) and 'soluble $N$-ethylmaleimide-sensitive factor (NSF) attachment proteins' (SNAP) as well as syntaxin-binding proteins (Sec1/Munc18) and synaptotagmins (Fig. 5a). Interestingly, syntaxins and syntaxin-interacting proteins enriched in the mouse PSD are involved in endocytic pathways, while those participating in presynaptic exocytosis were depleted from it (Fig. 5a), suggesting that the former are not biochemical contaminants. These included proteins with very well-established presynaptic functions, such as Snap25 or Vamp2. While these might be biochemical contaminants of the PSD preparation, several recent publications[30,31] have given evidence for their participation in postsynaptic processes. Thus, their localization in the PSD cannot be excluded. The mouse PSD was also specifically enriched in other complexes involved in endocytosis, including constituents of the 'endosomal-sorting complexes required for transport' (ESCRT) and 'homotypic fusion and vacuole protein sorting' (HOPS), and other key proteins for vesicle-mediated protein transport (Fig. 5b,c). To further explore this observation, we looked for these proteins in the PSD from human[5,32,33], rat[34–37] and an independent and recently generated mouse PSD proteome[38]. In all species we found more representatives of all these protein complexes than in zebrafish, with the exception of ESCRT components in rat (Supplementary Data 8).

To further investigate the depletion of some mammalian proteins from the zebrafish PSD, we asked whether the orthologous genes encoding these proteins were present in the zebrafish genome and, if so, whether they were expressed at low levels. Of the 745 mouse PSD proteins absent from zebrafish synaptic proteomes (SYN + PSD), 80% have orthologues in the zebrafish genome (Supplementary Fig. 9). To examine the possibility that those might be expressed at low levels in the zebrafish brain, we examined the 84 proteins shown in Fig. 5 using RNA sequencing data and found that the expression of most of these genes (63/84 = 75%) is low (less than 10 transcripts per million (TPM)) or very low (less than 1 TPM; Fig. 5 and Supplementary Fig. 10). The percentage of lowly expressed genes in this protein-depleted group is greater than what we see for all SYN- and PSD-encoding genes, where the percentage of detectable genes with TPM < 10 is 51% and 53%, respectively. This suggests that proteins depleted from the SYN and PSD of zebrafish show a corresponding low expression of encoding mRNA in the brain. Together, these findings indicate that both low levels of expression and absence of orthologues are mechanisms contributing to the depletion of synapse proteins from the zebrafish synapse.

To further test the hypothesis that mouse-specific PSD proteins added new functionalities to the PSD, we repeated the GO enrichment analysis with human[5,32,33] and rat[34–37] PSD proteins absent from the zebrafish synapse (Supplementary Data 7). For this extended analysis we also combined our mouse data with PSD proteins identified in other mouse studies[38–40]. We then looked for those terms significantly enriched in all species examined (Supplementary Data 7). We again found many enriched GO terms related to vesicle-mediated protein traffic, endocytosis and localization to the plasma membrane (Table 2). The rest of the enriched terms could be grouped into those related to actin filament organization, cation transport through the membrane and cell junctions involving the actin cytoskeleton (adherens junctions; Table 2). Finally, we looked for enriched KEGG pathways among PSD-specific proteins, as KEGG uses an annotation system different from that of GO. Again, we found that mammalian-specific PSD proteins are involved in endocytosis, regulation of actin cytoskeleton and adherens junctions among other pathways (Supplementary Data 7). Altogether, these analyses suggest that the zebrafish PSD has a reduced functional repertoire related to vesicle-mediated trafficking than that of mammals.

**A conserved vertebrate synapse proteome.** The comparison of vertebrate synapse proteomes from species separated from a common ancestor for over 400 million years provides an opportunity to identify the conserved elements within this highly complex structure, which will likely underpin the function of most vertebrates. We therefore sought to define the common set of vertebrate PSD proteins (vPSD) by identifying zebrafish PSD proteins with an orthologue in the mouse PSD. Accordingly, the vPSD consists of 1,101 proteins (Supplementary Data 9), including proteins from 12 major functional groups such as cytoskeletal proteins, ribosomal proteins, kinases, phosphatases, adenylate cyclase or small GTPases among others (Fig. 6a).

We next performed a set of analyses that show PSD protein sequences are remarkably conserved across vertebrates. First, SYN and PSD protein conservation was significantly higher than the average protein encoded in the genome in both species (median % of identity between zebrafish and mouse: for all zebrafish proteins, 49; SYN, 70; PSD, 72; Supplementary Fig. 11a; median % of protein identity between mouse and zebrafish for all mouse proteins, 49; SYN, 70; PSD, 70; Supplementary Fig. 11b). Second, we compared SYN and PSD protein conservation over ∼90 million years since humans and mice shared a common ancestor, and found higher identity in PSD compared to SYN proteins (Supplementary Fig. 11c) or in PSD-enriched proteins as compared to PSD-depleted ones (Supplementary Fig. 11d). Third, SYN and PSD were significantly more conserved than other

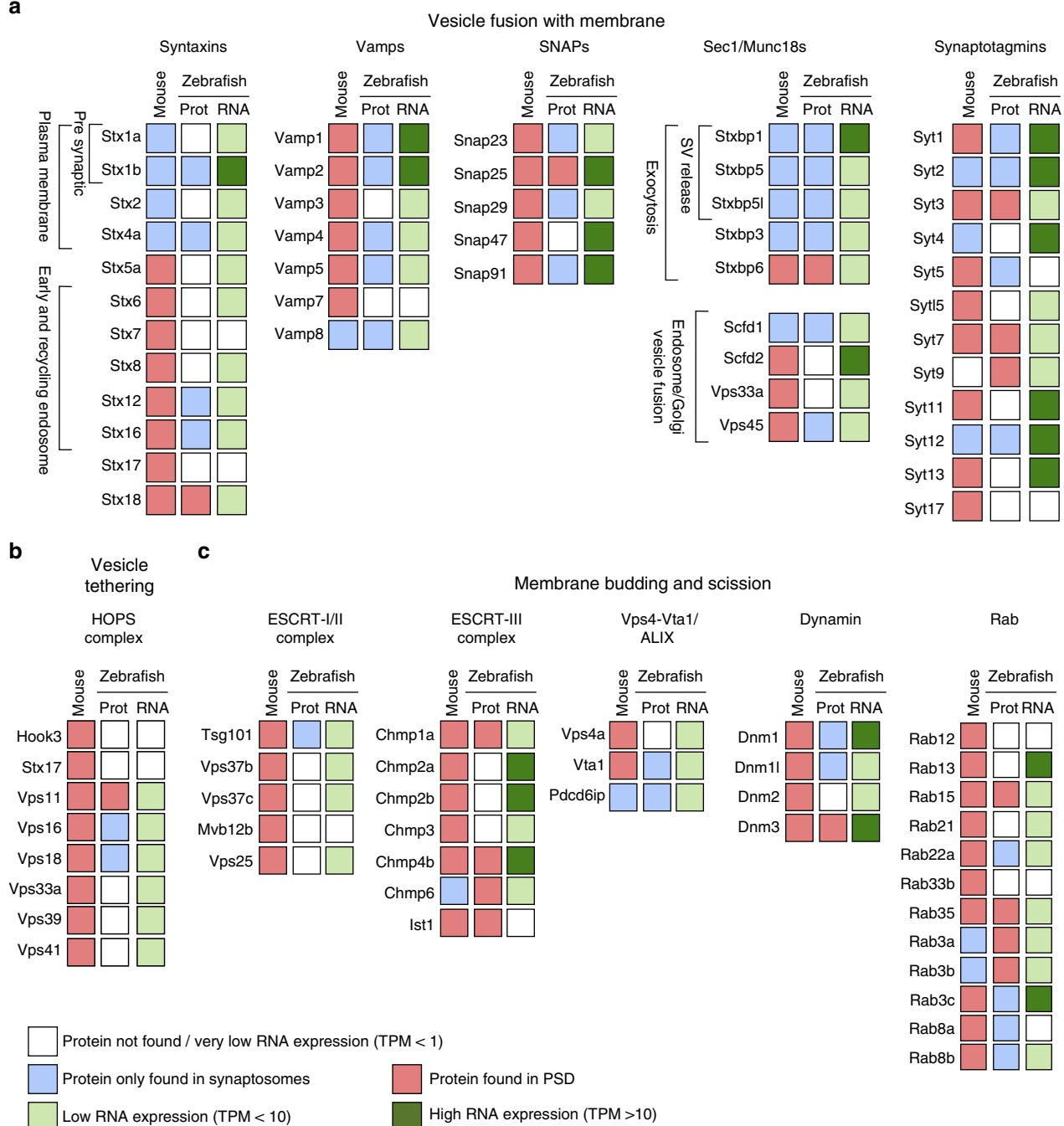

**Figure 5 | Differential expression of PSD proteins involved in intracellular vesicle biogenesis between mouse and zebrafish.** For each protein the mouse orthologue gene name is given. Proteomic data are provided for both species. For proteomic data, a white square denotes that we did not detect that protein, a blue square identifies proteins found in synaptosomes but not in PSDs and a red square identifies proteins found at the PSD. The average mRNA expression (mean transcripts per million (TPM) from four whole-brain biological replicates) was determined for each gene (for TPMs of individual samples, see Supplementary Fig. 10). For mRNA-sequencing data, a white box denotes a mean expression of <1 TPM, a pale green box <10 TPM and dark green >10 TPM. (**a**) Proteins involved in vesicle fusion with membranes, including SNARE complex proteins: Syntaxins, Vamps and SNAPs, syntaxin-binding proteins (Sec1/Munc18) and Synaptotagmins. (**b**) Proteins form the HOPS complex involved in membrane vesicle tethering to membranes. (**c**) Proteins involved in membrane bending/budding and scission to form vesicles, including proteins from ESCRT complexes (ESCRT I, II, III and Vps4-Vta1/ALIX), Dynamins and small GTPases from the Rab family.

proteins expressed in the brain (Supplementary Fig. 11e–h), which already exhibit high conservation[41]. Fourth, the vPSD showed a greater level of conservation than the entire PSD (Fig. 6b,c). This observation held true when the vPSD was obtained by comparing zebrafish PSD proteins with human[5,32,33], mouse[38–40] and rat[34–37] PSD proteomes (median vPSD protein identity 66% and median mammalian-specific PSD protein identity 61.5%; significantly different, Mann–Whitney U-test, P < 0.0001). Fifth, we asked whether the species-specific PSD (Zf-sPSD and Mm-sPSD) proteins also showed this high conservation, and found that they were significantly lower, even when compared with whole-brain proteomes (Fig. 6b,c). To

**Table 2 | Functional annotation of mammalian PSD proteins absent from the zebrafish synapse.**

| | Human | | Rat | | Mouse | |
|---|---|---|---|---|---|---|
| | Fold change* | P value# | Fold change* | P value# | Fold change* | P value# |
| *Related to vesicle-mediated protein traffic and localization* | | | | | | |
| Regulation of vesicle-mediated transport (GO:0060627) | 3.3 | 1.3E − 04 | 3.4 | 1.7E − 03 | 2.3 | 3.0E − 04 |
| Vesicle-mediated transport (GO:0016192) | 2.9 | 5.8E − 10 | 2.4 | 7.1E − 03 | 2.2 | 1.7E − 12 |
| Cytoplasmic transport (GO:0016482) | 2.9 | 5.0E − 06 | 2.6 | 1.9E − 02 | 2.0 | 1.2E − 04 |
| Establishment of protein localization (GO:0045184) | 2.9 | 1.2E − 12 | 2.2 | 1.7E − 03 | 2.2 | 2.5E − 15 |
| Protein transport (GO:0015031) | 2.8 | 2.7E − 10 | 2.3 | 4.1E − 03 | 2.2 | 3.5E − 14 |
| Intracellular transport (GO:0046907) | 2.7 | 9.9E − 10 | 2.5 | 6.3E − 06 | 2.2 | 4.0E − 15 |
| Endosome (GO:0005768) | 2.2 | 3.3E − 02 | 2.5 | 3.2E − 03 | 2.3 | 6.8E − 10 |
| *Related to protein location at plasma membrane* | | | | | | |
| Extrinsic component cytoplasmic side of plasma membrane (GO:0031234) | 5.8 | 2.8E − 04 | 5.9 | 5.2E − 03 | 3.5 | 9.4E − 04 |
| Extrinsic component of plasma membrane (GO:0019897) | 5.1 | 1.1E − 04 | 5.3 | 8.1E − 04 | 3.3 | 6.5E − 05 |
| Cytoplasmic side of plasma membrane (GO:0009898) | 4.5 | 5.3E − 04 | 6.1 | 2.6E − 06 | 3.7 | 1.1E − 07 |
| Cytoplasmic side of membrane (GO:0098562) | 4.2 | 1.4E − 03 | 5.7 | 7.7E − 06 | 3.5 | 2.1E − 07 |
| Extrinsic component of membrane (GO:0019898) | 3.3 | 6.3E − 03 | 3.6 | 1.1E − 02 | 2.5 | 2.8E − 04 |
| *Related to actin filaments organization* | | | | | | |
| Regulation of actin filament length (GO:0030832) | 4.6 | 2.8E − 03 | 4.8 | 1.7E − 02 | 2.9 | 9.7E − 03 |
| Actin cytoskeleton organization (GO:0030036) | 4.0 | 1.8E − 08 | 3.3 | 3.7E − 03 | 2.4 | 7.9E − 06 |
| Actin filament-based process (GO:0030029) | 4.0 | 1.3E − 09 | 3.4 | 5.9E − 04 | 2.6 | 3.8E − 08 |
| Regulation of cytoskeleton organization (GO:0051493) | 3.4 | 8.1E − 06 | 3.2 | 3.9E − 03 | 2.7 | 3.0E − 09 |
| *Related to cation transport* | | | | | | |
| Regulation of cation transmembrane transport (GO:1904062) | 4.4 | 1.1E − 04 | 4.6 | 1.2E − 03 | 2.7 | 2.7E − 03 |
| Regulation of metal ion transport (GO:0010959) | 3.0 | 2.3E − 02 | 3.4 | 1.7E − 02 | 2.2 | 4.2E − 03 |
| Transmembrane transporter complex (GO:1902495) | 3.0 | 2.8E − 02 | 3.4 | 1.1E − 02 | 2.7 | 5.0E − 06 |
| Transporter complex (GO:1990351) | 2.9 | 3.8E − 02 | 3.3 | 1.5E − 02 | 2.7 | 1.2E − 06 |
| *Related to cell junction* | | | | | | |
| Adherens junction (GO:0005912) | 3.9 | 1.8E − 09 | 3.3 | 2.1E − 04 | 2.2 | 1.3E − 04 |
| Cell–substrate junction (GO:0030055) | 3.9 | 1.1E − 07 | 3.7 | 3.8E − 05 | 2.0 | 1.7E − 02 |
| Anchoring junction (GO:0070161) | 3.7 | 6.3E − 09 | 3.3 | 1.3E − 04 | 2.2 | 9.8E − 05 |
| Cell junction (GO:0030054) | 2.9 | 2.0E − 13 | 3.0 | 3.0E − 12 | 2.4 | 2.1E − 22 |

GO, Gene Ontology; PSD, postsynaptic density.
*Fold enrichment of observed proteins per GO term.
#Binomial statistics P value.

further validate these findings we took PSD proteins specific to each species and eliminated those found in the SYN fraction of the other species. Again, vPSD proteins showed higher percentages of protein identity than Zf-sPSD and Mm-sPSD (Fig. 6d). Thus, the vPSD is a highly conserved set of ∼1,000 proteins common to vertebrate species that shared an ancestor ∼450 million years ago.

## Discussion

Our study of zebrafish synapse proteomes has led to a number of new insights into the evolution of synapses. First, retention of duplicated synapse genes following the TSGD has generated an increase in molecular complexity in zebrafish. Second, despite this increase in proteome size, the PSD complexity was lower in zebrafish than in mammals. Third, the characterization of a conserved vPSD indicates that high molecular complexity is a core feature across bony fish, amphibians, reptiles, birds and mammals. Fourth, lineage-specific changes in proteins around this vPSD result in species-specific differences in synapse composition.

Our data show that vertebrate synapse proteomes have been shaped by multiple WGDs[11,42] including the TSGD around 300 million years ago[13] and two WGDs 150 million years earlier. Following a WGD, most duplicated genes accumulate deleterious mutations becoming pseudogenes[43] so that only a small number of

the originally duplicated genes are retained. For instance, the last update of the zebrafish genome identifies 3,440 gene pairs (ohnologues) remaining from the TSGD[14], representing a retention of ∼25% of the novel duplicates. Nevertheless, when considering the number of paralogues found in zebrafish synaptic protein families we found a significant increase compared with mouse, indicating that after the TSGD many synaptic genes have been retained in the zebrafish genome. Indeed, we have shown that zebrafish genes expressed in synapses have been retained more frequently after the TSGD than other coding genes in the genome or other genes expressed in the brain, suggesting their functional importance. Consistent with this, studies reporting the types of genes retained in vertebrates after the two rounds of WGD show that among those more commonly retained are genes involved in synaptic function[44]. Our data support the idea that genes performing synaptic functions are retained at higher frequencies following successive rounds of WGD. This is consistent with the view that their sub- and/or neo-functionalization[45] expanded synaptic molecular complexity and diversity, contributing to improved fitness.

Consistent with the conservation of the vPSD we found that many ultrastructural features of the postsynaptic density observed in mammals were found in zebrafish. Asymmetric synapses (containing PSDs) in olfactory bulb and telencephalon were particularly similar to those observed in mammals. However, some of the synapses identified in the optic tectum and

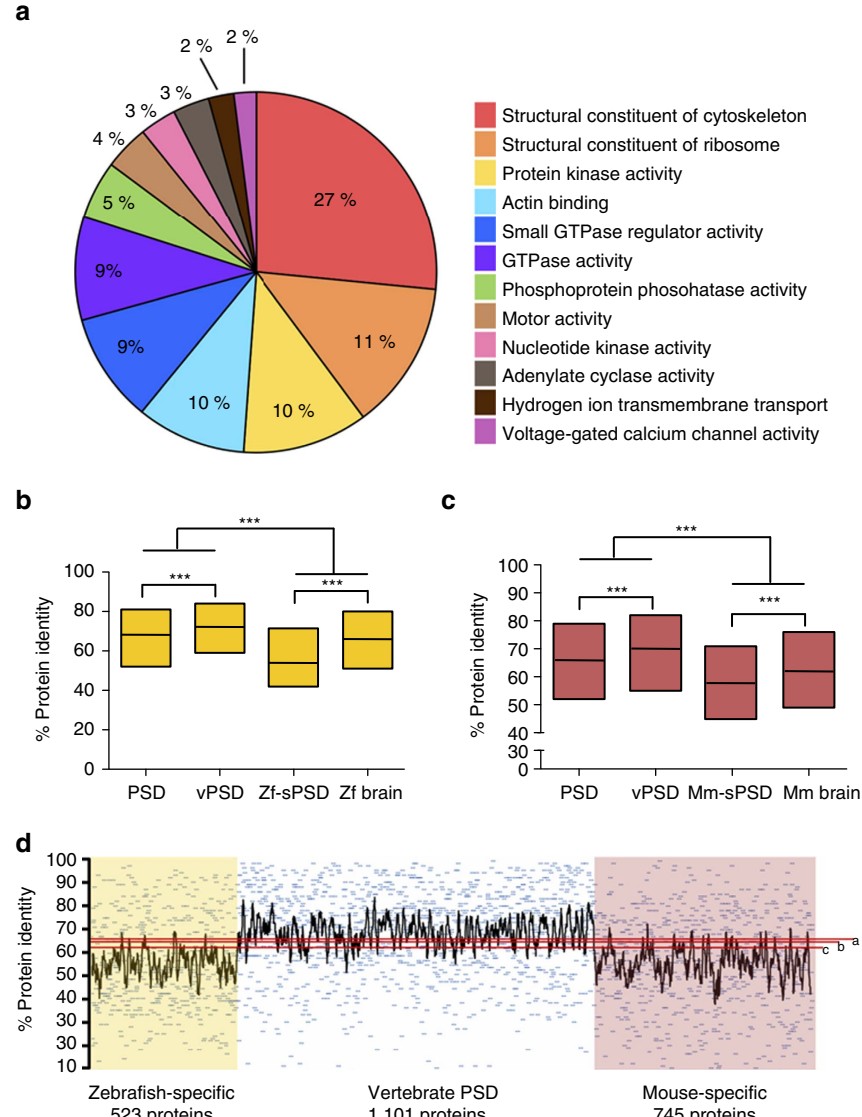

**Figure 6 | Conservation of the core vertebrate PSD machinery.** (**a**) Pie chart of main GO-Slim Molecular Function categories enriched among vPSD proteins. (**b**) Box plots for the percentage of protein identity since the last common ancestor between zebrafish and mouse of zebrafish PSD proteins, PSD components found in both species (vPSD), Zf-sPSD and a zebrafish brain proteome[29]. Distributions compared using the Mann–Whitney $U$-test (***$P < 0.001$). (**c**) Box plots for the percentage of protein identity since the last common ancestor between zebrafish and mouse of mouse PSD proteins, PSD components found in both species (vPSD), Mm-sPSD and a mouse brain proteome[26]. Distributions compared using the Mann–Whitney $U$-test (***$P < 0.001$). (**d**) Percentage of protein identity for individual proteins found only in zebrafish PSD (yellow), in both species PSD (white) and only in mouse PSD (red). Red lines represent the median percentage of protein identity for two zebrafish (a (ref. 29) and b (ref. 28)) and a mouse (c (ref. 26)) whole-brain proteomes.

cerebellum presented particular morphologies. These have also been reported in other bony fish species[18,19,22], further supporting synaptic diversity across vertebrates. Particularly remarkable were asymmetric synapses from the optic tectum, which show a very clear PSD but not an equally obvious dendritic spine, as the presynaptic boutons contacted thin structures that might correspond with dendritic shafts. Similar observations have been made in a few other teleost species[18,19]. Yet, in the superior colliculus of mammals, the homologous brain region to the fish optic tectum, asymmetric synapses are mainly formed on dendritic spines[46]. Since previous studies in mice show that synapse proteins appearing after the two WGDs contributed to synaptic diversity[42], future neuroanatomical studies could determine whether synapse proteins arising from the TSGD are allocated into different individual synapses in the zebrafish brain.

We unexpectedly found several lines of evidence that highlight the specialization of the synapse proteome. While studying the frequency of new domains found per protein, we observed a strong increase in the PSD and synaptosome proteins (in both mouse and zebrafish) compared with whole-brain proteomes or all protein-coding genes in the genome. Second, in previous work we have reported that PSD proteins have been subjected to very high levels of sequence conservation during mammalian evolution[5,39], and the present study indicates that this evolutionary constraint has occurred throughout vertebrate evolution, and not only for postsynaptic proteins, but for synaptic molecules overall. This high conservation suggests that the proteins are important for fitness, and consistent with this, disease-causing mutations have been documented in several hundred different genes encoding the human postsynaptic proteome. We suggest that

the vPSD data set will be particularly valuable for future human genetic studies and behavioural genetic screens in zebrafish.

The identification of mammalian-specific PSD proteins, which were absent from the zebrafish synapse, opens new lines of investigation into the mammalian brain. No relevant differences were observed when comparing human and mouse postsynaptic proteomes[39], suggesting that it may be a key difference between mammals and fish. Our observations that there were fewer orthologues and low levels of mRNA expression suggest that gene loss and transcriptional regulatory changes are contributing mechanisms. In addition, there may be post-translational mechanisms such as protein stability. Although we cannot fully exclude the contribution of technical reasons, the observed enrichment in particular synaptic functions in the set of mammalian-specific PSD proteins suggests that there are biologically relevant differences between the zebrafish and mouse PSD. Particularly noticeable was the high number of SNARE complex components (Syntaxins, Vamps and SNAPs) and associated proteins (Sec1/Munc18s and Synaptotagmins). Importantly, only Syntaxins and Sec1/Munc18s proteins with a clear role in endocytosis[47] were found in the mouse PSD, and several of these (Syntaxin 12 (ref. 48), SNAP23 (ref. 49) or SNAP47 (ref. 50)) play a role in AMPA receptor trafficking. In addition, HOPS and ESCRT complexes, which also participate in the endocytic machinery[51,52], were also found enriched among Mm-sPSD proteins and are also found in the PSD of other mammalian species. A recent study shows that some ESCRT components are at the mouse PSD, where they contribute to the regulation of synaptic plasticity and confer specific structural characteristics to the postsynaptic membrane[53]. To extend and validate the analysis of Mm-sPSD we repeated the analysis with human and rat and confirmed that proteins incorporated into the PSD after the fish divergence added functionality related to vesicle-mediated protein traffic, protein location to the plasma membrane and actin filament organization as well as the regulation of cation transport and establishment of adherens junctions.

Our findings have several implications for the use of zebrafish as models of human brain disease. Zebrafish are used to model neurodegeneration[15], depression[54], autism[17] and schizophrenia[55] among others[56] and for neuropharmacological[16] and neurotoxicology[57] research. These disorders and interventions directly and indirectly influence synapse protein structure and function. Therefore, it is potentially important to consider the following issues: the additional zebrafish-specific paralogues arising from the TSGD will increase redundancy and potentially mask phenotypes in mutations within that gene family. The additional paralogues may also have undergone species-specific neofunctionalization resulting in species-specific phenotypes. The species-specific differences in overall complexity alters many classes of proteins at multiple levels of signalling pathways, and therefore the postsynaptic signalling networks will have a different structure, potentially resulting in differential robustness and signalling capacity. It is also interesting to consider the finding that the vPSD is highly conserved, and perhaps this subset of the synapse proteome will be preferred when modelling human mutations. While these considerations may be important for studies aimed at modelling or treating human diseases, we also wish to highlight that these differences will be of interest in the study of fundamental synaptic physiology and behaviour of zebrafish. The demonstration of paralogue-specific behavioural functions in mice and conserved phenotypes in humans and mice[2,3] illustrate that the synapse proteome complexity of zebrafish will be a major factor in their behavioural repertoire.

Synapse proteome data from mice and humans have been used in a wide range of applications. For example, the mammalian data have been used in many human genetic studies including those showing that schizophrenia is primarily a synaptic disorder where multiple susceptibility genes converge on the PSD[10,58,59]. Mouse proteome data have been used to show that the mRNAs that interact with the Fragile X Mental Retardation Protein predominantly target the PSD[60]. PSD proteome data were used to study the behavioural and physiological phenotypes controlled by synapses using the Mouse and Human phenotype ontologies[5]. Thus, we expect that the zebrafish synapse proteome data will be a valuable resource that can be exploited with many orthogonal data sets and technical approaches. All data and tables from this study are freely available through the Genes to Cognition database (http://www.genes2cognition.org/publications/zebrafish-prot/).

## Methods

**Ethics statement.** Mouse (*Mus musculus*) and zebrafish (*Danio rerio*) were treated in accordance with the British Home Office regulations (Animal Scientific Procedures Act, 1986; Project Licence PPL80/2,337 to Professor Seth Grant). Animal protocols were approved by the local ethical committee on animal experimentation at the Wellcome Trust Sanger Institute. Animals were housed in The Wellcome Trust Sanger Institute animal facility.

**Zebrafish and mouse brain samples.** We used whole-brain samples dissected from male and female adult *D. rerio* and 6–8-week-old mice from the 129 Strain. After dissection brain tissue was immediately frozen in liquid nitrogen and stored at −80 °C until being used for extraction of synaptosomes and PSDs. Zebrafish were from the following strains: H Longfin, Tubingen Longfin, Tubingen, AB, WIK, LON and SAT. Before dissection, fish were killed by an overdose of the fish anaesthetic Tricane at 0.1% (w/v) and mice were killed by cervical dislocation. Three independent biological replicas were prepared for both zebrafish and mouse brain samples, and each contained 0.7–1 g of tissue. All mouse and zebrafish sample-processing steps were performed in parallel and peptide fractions from all samples analysed back-to-back by mass spectrometry within just over a week. Performance and sensitivity were monitored throughout. Sample size was established based on the standard in the field. No method was used to randomize animals between experimental groups; neither investigators were blinded to the species origin of each sample.

**Electron microscopy.** Electron microscopy was performed with the brain from two adult specimens. In each case the following brain regions were studied: olfactory bulb, telencephalon, optic tectum and cerebellum. Zebrafish brains were dissected under cold primary fixative containing 2% paraformaldehyde and 2.5% glutaraldehyde in 0.1 M sodium cacodylate buffer at pH 7.42. Each brain was halved in mid-sagittal section and then separated in transverse sections into telencephalon (including olfactory bulb), optic tectum, cerebellum and medulla. These four compartments were fixed for the remainder of 2 h, rinsed and post-fixed in 1% osmium tetroxide for an hour, mordanted with 1% tannic acid and dehydrated in an ethanol series, en bloc staining with 2% uranyl acetate at the 30% stage. Following immersion in propylene oxide, the brain segments were embedded in TAAB 812 resin. Semi-thin sections (0.5 μm) were cut on a Leica UCT ultra-microtome and stained with toluidine blue on a microscope slide. Images were recorded on a Zeiss Axiovert CCD (charge-coupled device) camera and areas selected for 50 nm ultrathin sectioning. Thin sections were collected on copper/palladium grids and contrasted with uranyl acetate and lead citrate before viewing on an FEI 120kV Spirit Biotwin TEM and recording CCD images on an F4.15 Tietz camera. PSD lengths and areas were measured on electron microscopy images with the FiJi image-processing package[61]. Groups were compared through the median lengths and areas, and significant differences were analysed through the Kruskal–Wallis non-parametric test. All analyses were performed with the SPSS statistics software (IBM).

**Isolation and characterization of synaptosomes and PSDs.** Mouse (*M. musculus*) and zebrafish (*D. rerio*) samples were fractionated in parallel, using previously reported methods[39]. Briefly, ∼1 g of whole-brain tissue was homogenized 9:1 (v:w) using a glass–teflon tissue grinder in a buffer containing Tris 50 mM, pH 7.4, 0.3 M sucrose, 5 mM EDTA and the protease inhibitors 1 mM phenylmethylsulphonyl fluoride (PMSF), 2 μM Aprotinin and 2 μM Leupeptin. The homogenate was centrifuged at 800g to pellet nuclei and cell debris; the resulting supernatant was then centrifuged at 16,000g and the pellet was resuspended 5:1(v:w) in Tris 50 mM, pH 8.1, 5 mM EDTA, 1 mM PMSF, 2 μM Aprotinin and 2 μM Leupeptin, and chilled in ice for 45 min. Sucrose was added to a final 34% (w/w) concentration. A sucrose gradient was prepared with equal volumes of the following layers (bottom to top): sample, Tris 50 mM, pH 7.4, 0.85 M sucrose and Tris 50 mM, pH 7.4, 0.3 M sucrose. This gradient was then

ultracentrifuged for 2 h at 60,000g and the interphase between 34 and 28.5% sucrose was collected, diluted to 10% sucrose with Tris 50 mM, pH 7.4 and centrifuged again at 48,000g during 30 min. Pellet was resuspended in 1 ml of Tris 50 mM, pH 7.4 to generate the SYN fraction. A range of 5–10% of this solution was set apart and later used for proteomics profiling. To obtain the final PSD fraction the remaining synapstosomal fraction was mixed with an equal volume of 3% Triton X-100 and chilled in ice for 30 min. Sample was finally layered on top of 10 ml of Tris 50 mM, pH 7.4, 0.85 M sucrose and centrifuged at 104,000g for 1 h to produce a pellet containing the PSD fractions that are solubilized in Tris 50 mM, pH 7.41% SDS. Enrichment of postsynaptic proteins in postsynaptic density fractions was assessed by immunoblotting using the postsynaptic marker protein PSD95 (antibody used: Affinity, ref. MA1-045).

**Mass spectrometry-based proteomics.** In-gel digestion was performed as reported previously[5]. Extracted peptides (six fractions per sample) were analysed using nanoLC-MS/MS on a LTQ-Orbitrap Velos (Thermo Fisher) hybrid mass spectrometer equipped with a nanospray source, coupled with an Ultimate 3000 Nano/Capillary LC System (Dionex). The system was controlled with Xcalibur 2.1 (Thermo Fisher) and DCMSLink 2.08 (Dionex). Peptides were desalted on-line using a micro-Precolumn cartridge (C18 Pepmap 100, LC Packings) and then separated using a 120 min reverse phase gradient (4–32% acetonitrile/0.1% formic acid) on an EASY-Spray column, 50 cm × 75 µm ID, PepMap C18, 2 µm particles, 100 Å pore size (Thermo). The LTQ-Orbitrap Velos was operated with a cycle of one MS (in the Orbitrap) acquired at a resolution of 60,000 at m/z 400, with the top 10 most abundant multiply charged (2 + and higher) ions in a given chromatographic window subjected to MS/MS fragmentation in the linear ion trap. An FTMS target values of 1e6 and an ion trap MSn target value of 5e3 was used and with the lock mass (445.120025) enabled. Maximum FTMS scan accumulation time of 150 ms and maximum ion trap MSn scan accumulation time of 100 ms was used. Dynamic exclusion was enabled with a repeat duration of 45 s with an exclusion list of 500 and exclusion duration of 30 s.

MS data were analysed using MaxQuant[62] version 1.5.2.8. Data were searched against mouse (GRCm38.p3 (GCA_000001635.5)) or zebrafish GRCz10 (GCA_000002035.3) UniProt sequence databases (downloaded June 2015) using the following search parameters: trypsin with a maximum of two missed cleavages, 7 p.p.m. for MS mass tolerance, 0.5 Da for MS/MS mass tolerance, with acetyl (protein N-term) and oxidation (M) set as variable modifications and carbamidomethyl (C) as a fixed modification. A protein false discovery rate (FDR) of 0.01 and a peptide FDR of 0.01 were used for identification level cutoffs. Variance in protein abundance data was similar within species replicas and between species. In addition, for a protein to be included in the final set of SYN or PSD proteins it had to be identified with at least one unique peptide in each of the three SYN or PSD replicas. Label-free quantification (LFQ) was performed using MaxQuant LFQ intensities[63], and statistical analysis was performed using Perseus[64] as follows. The data set was filtered to remove proteins with less than two valid LFQ values in at least one group (PSD or SYN). LFQ intensities were log2-transformed and missing values were imputed using a downshifted normal distribution (width 0.3, downshift 1.8). Next t-testing was performed with correction for multiple hypothesis testing using a permutation-based FDR of 0.05.

**Gene homology.** Gene orthology relationships between zebrafish and mouse and percentage of protein sequence identity were taken from Ensembl database[65] version 81, containing the last update of the zebrafish genome[14]. Statistical comparison of protein identities between different proteomic sets was performed using the Mann–Whitney U-test.

**Functional classification of synaptic proteins.** For the functional classification with high-level categories, Ensembl mouse identifiers for mouse proteins or orthologous mouse identifiers for zebrafish proteins were integrated with functional annotation from the Ingenuity knowledgebase[66], using IPA (QIAGEN Redwood City www.qiagen.com/ingenuity). Information of predicted cellular localization (cytoplasm, extracellular space, nucleus, plasma membrane and other) and IPA protein types (cytokine, enzyme, G-protein coupled receptor (GPCR), ion channel, kinase, peptidase, phosphatase, transcription regulator, translation regulator, transporter and other) were obtained. Counts, comparisons and plots of proteins within each species and category were conducted using R. To reproducibly call orthologous sequences between species for a large data set, the Ensembl biomart database was queried using the bioconductor package biomaRt[67]. All orthologues were obtained and counted in each species to determine orthology type, either 1:1, 1:many, many:1, many:many or unique to a species (no orthologue known). For the analysis of all protein families, we used Ensembl identifiers of mouse, zebrafish and mouse orthologues of zebrafish proteins to retrieve Ensembl Protein Families from Ensembl database[65] version 81, containing the last update of the zebrafish genome[14]. Families with an unknown function were not considered.

**Domain analysis.** Protein domain composition for all genes in the mouse and zebrafish genome data sets were obtained via biomart; subsets corresponding to brain, SYN and PSD data sets were obtained from this single data set. The total counts for domains for each protein were determined, and the unique protein types were determined by removing duplicate domains within any single protein. The complexity of the proteome was calculated by comparing the cumulative frequency of unique domains per protein within a proteome. Distributions were compared using a two-tailed Kolmogorov–Smirnov test applied to cumulative frequency distributions. All statistical calculations were conducted in R.

**GO enrichment analysis.** Zebrafish and mouse synaptic proteins were annotated for 'Cellular Component' and 'Biological Process' gene ontology[68] terms using the Panther database and analysis tools[69]. Binomial statistics were used to compare GO term over-representation using the whole genome as the background set, and the Bonferroni test was used to correct for multiple testing. To account for the lower level of GO annotations found in zebrafish, Zf-sPSD proteins were searched for enrichment against the zebrafish and mouse genomes. Terms found enriched in both species were not further considered. Equivalent enrichment analysis was also performed with categories from the KEGG pathway database[70].

**Analysis of protein sequence identity.** Percentage of protein sequence identity between mouse and zebrafish or mouse and human proteins was taken from Ensembl database[65] version 81. Differences between protein sequence identity were calculated with Mann–Whitney U-test.

**RNA sequencing.** Four whole brains were removed and placed in RNAlater before RNA isolated using the Qiagen RNeasy Plus Mini Kit, and 150 bp paired end Illumina sequencing was conducted at Barts and the London Genome Centre. Adapters were removed from the raw reads using Cutadapt. TopHat2 was used as a wrapper for the alignment programme Bowtie2 to map sequence reads to the reference genome (Danio_rerio.GRCz10.86 obtained via Ensembl), reads were converted into counts using HTSeq and converted to TPM. For comparison of each gene, the average expression (mean TPM from four whole-brain biological replicates) was determined for each gene. Where multiple transcripts for a given gene are known, these were combined to result in a single mean TPM per gene.

**Data availability.** We have constructed a freely available database and web resource that includes links to a wide variety of biomedical data sets: (http://www.genes2cognition.org/publications/zebrafish/).

Mass spectrometry proteomics data have been deposited to the ProteomeXchange Consortium via the PRIDE[71] partner repository with the data set identifier PXD005630.

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

## Acknowledgements

We thank D.L. Stemple, K. Howe and K. Elsegood for technical support; L. Chakrabarti and K. Sharma YVR for isolation of zebrafish RNA; M.D. Croning and J. Menedez Montes for web development; R. Lujan (Universidad Castilla-La Mancha, Albacete, Spain) for critical review of electron microscopy data; I. Gich (IIB Sant Pau) for support in biostatistics; and D. Maizels for artwork. This study was supported by AB, Spanish grants ref. BFU2012-34398 and BFU2015-69717-P, Career Integration Grant, ref. 304111, Marie Curie Intra-European Fellowship, ref. 221540, Ramón y Cajal Fellowship, ref. RYC-2011-08391p; M.O.C. was supported by the Royal Society (R/144823-11:1); J.S.C. and S.G.N.G. was supported by Wellcome Trust; R.D.E. was supported by the University of Nottingham Advanced Data Analysis Centre. A.I. was supported by an international PhD studentship from Consejo Nacional de Ciencia y Tecnologia (CONACYT) Mexico.

## Author contributions

Conceptualization: À.B. and S.G.N.G.; formal analysis: R.R.-V. and R.D.E.; investigation: À.B., M.O.C., R.R.-V., G.G., D.G., A.I. and R.D.E.; resources: J.S.C.; writing—original

draft: À.B., R.D.E. and S.G.N.G.; writing—review and editing: À.B., M.O.C., R.R.-V., G.G., R.D.E. and S.G.N.G.; supervision: À.B. and S.G.N.G.; project administration: À.B. and S.G.N.G.; funding acquisition: À.B. and S.G.N.G.

## Additional information

**Competing financial interests:** The authors declare no competing financial interests.

**How to cite this article**: Bayés, À. *et al.* Evolution of complexity in the zebrafish synapse proteome. *Nat. Commun.* **8**, 14613 doi: 10.1038/ncomms14613 (2017).

**Publisher's note**: 

