## [Peer Review File · Nature Communications]

Reviewers' comments:

Reviewer #1 (Remarks to the Author):

Bayes et al present a comparative analysis of mouse and zebrafish synaptic proteomes. They find that the zebrafish PSD proteome is of smaller size than that found in mammals. The lower complexity of zebrafish PSDs results from fewer protein families. Interestingly, a major difference between mouse and zebrafish PSD proteomes is an enrichment in proteins involved in endocytosis, the regulation of vesicle transport, the actin cytoskeleton, and adherens junctions, in mouse PSDs. Together, these findings suggest that zebrafish synaptic proteomes have a reduced functional repertoire compared to the mammalian proteomes.

Overall, the data presented in this paper and made available in the genes2cognition database are a good reference database for zebrafish researchers and a valuable resource to the field. It is the first paper describing the zebrafish synaptic and PSD proteome, allowing a better understanding of vertebrate PSD complexity and evolution. The bioinformatics approach and methodology used seem sound. The manuscript is clearly written. The paper is essentially descriptive in nature: zebrafish synapses are studied at the ultrastructural and proteome level. Its main conclusion, that zebrafish PSDs have a reduced functional repertoire compared to their mammalian counterpart, is not functionally tested. However, this is not a trivial thing to do and I would support publication of this paper based on its merit as a resource.

Comments:

1) The authors mention that proteomic profiling of zebrafish and mouse synaptic proteomes was performed in triplicate. Please clarify in the text whether all brains of mouse or zebrafish were pooled and split into 3 identical samples or whether the whole experiment was performed 3 times independently.

2) A large difference was observed in the number of significantly enriched terms between mouse and zebrafish PSD proteomes: mouse has more GO terms than zebrafish. Is it possible that the mouse GO annotations database is more complete and detailed than the zebrafish one and that this may have influenced the findings? Please comment.

3) The main finding that mouse PSDs are enriched in SNARE, ESCRT and HOPS proteins compared to zebrafish PSDs is interesting and could be further explored. How do these findings compare to rat and human PSDs? Supp Table 7 lists GO enrichment analysis with human and rat proteins, but it is not easy to cross compare the findings with zebrafish. Can these results be shown in a table side by side with the zebrafish data? Independent validation of these findings, using for instance western blot, would enhance these findings.

4) Can the authors comment in the Discussion on the implications of their findings for the use of zebrafish as a model of human synaptopathies? Their findings suggest that a main difference between mammalian and zebrafish synaptic proteomes may be that the former have a much more complex repertoire of proteins involved in regulated protein trafficking. How does this impact the use of zebrafish as a model in human brain disorder, neuropharmacology and neurotoxicology research, as outlined in the opening paragraph of the Introduction?

Reviewer #2 (Remarks to the Author):

This study aims to characterize the zebrafish synapse/PSD structures and their proteomes. Comparison to mouse PSD proteome suggests a lower complexity of zebrafish PSD with fewer protein families comprising of more gene duplicates. The authors have performed extensive EM and quantitative proteomics experiments. The approaches and the subsequent analyses are well established. The conclusion and the references are appropriate. I am not aware of any major fault in this study. However, this study is purely descriptive. It does not provide useful data that advance our understanding of zebrafish behaviour or the use of zebrafish as a model for animal behavioural/brain disorder studies.

There are several minor points that deserve discussion.

The authors highlight the lower complexity of zebrafish PSD proteome. I would define PSD proteins as proteins either unique or enriched in PSD (compared to synaptosome). As such, zebrafish PSD proteome contains only 10% less proteins. Is it still significantly different from mouse PSD? It should also be realized that the comparison of protein abundance levels may be hampered by the biochemical challenges in extracting the same synapse material from the vastly different zebrafish and mouse brains.

Figure 5 shows the presence of classic presynaptic exocytosis proteins in PSD. I suppose Stx1-Snap25-Vamp2 were highly depleted in PSD? These proteins are generally not considered as PSD proteins, and should not be highlighted as PSD proteins.

Reviewer #3 (Remarks to the Author):

The manuscript entitled, "The zebrafish synapse proteome: A resource for evolutionary and comparative analysis of vertebrate brain complexity," provides a detailed comparison of the proteome from two different synapse preparations and between zebrafish and mouse. The manuscript includes ultrastructural and morphological analyses of synapses from a limited number of brain regions.

The sample preparation, proteomics and statistical analysis follow well-established protocols established by this group. Overall this is a very interesting paper that would provide considerable information for future studies in synapse structure/function relationships. The data also has the potential to broaden the fields understanding of a minimal functional unit of a vertebrate synapse and modifications of this unit in different lineages.

The relative merit of the proteomics remains high with only minor clarifications needed:

The notion that the fish synaptosome preparation contains fewer proteins is reasonable based on the data and interesting considering the selective retention of synaptic proteins following the GWD. Differences may also represent species specific utilization. But are there other explanations to consider. First, the homogenization and centrifugation methods used for the preparations is a standard protocol developed for mammalian tissues. Is the method be equally effective on a fish brain? The absence of a specific gene may be a technical issue not a biological difference. Does the author have the ability to access the quality of the tissue homogenates from the fish beyond the one immunoblot provided? This is especially important considering the considerable lack proteins associated with vesicle trafficking and endocytosis which one might expect to be highly conserved.

Secondly, along this line, the presence or absence of specific genes in zebrafish, presented in Fig 5 seems incomplete. A quick search of ensembl shows annotated sequences for the majority of the syntaxins and vsp's, and the zebrafish information server provides in situ hybridization data for many of these genes, yet the boxes are white indicating not in fish. Is this a misreading by this reviewer,

please clarify.

Once clarified, can the authors elaborate on the relative importance of neofunctionalization in mammals versus gene loss in fish.

In their analysis did the authors consider tandem duplication versus the GWD as a source of additional genes.

Major concerns which need be addressed.

The ultrastructural data, beautiful and very clearly presented, is not tied into the major points of the manuscript. A link between the differences in distribution of the types of synapses observed and potential differences in proteomics profiles is eluded to and would have been interesting as a follow-up, but these cannot be made since the whole brain was used in the proteomics. Any variation in protein expression within or across species or association with a specific class of synapses are lost.

Although this may be correct that these are the first EM analysis of these brain regions in an adult zebrafish, the authors failed to cite any relevant work from other species. A quick web search popped up numerous examples of ultrastructural analysis from the 1970's and 80's of a considerable many regions of the teleost brain included those in this manuscript, with descriptions of synapse types, locations on various cell structures, morphometric analysis and variations across strata. Please cite appropriate published work.

In the methods it seems a single animal was used for the morphological analysis which limits the ability to draw large conclusions from the sample. Better description and possibly images of the low magnification images indicating the precise region used for the ultrastructure is needed so as not to hinder a readers ability to replicate the findings.

Supplemental methods states, "Mean lengths and areas were compared by an analysis of variance (ANOVA) test and differences between groups were identified through the Tukey post hoc test. Significant differences between groups were checked with Kruskal-Wallis non-parametric test." There are several problems with this statement. First, an ANOVA followed by Tukey are parametric analysis that makes several assumption such as the data are normally distributed. KW in a non-parametric analysis. The authors must decide which type of test is allowed by the experimental design. Secondly analysis by two different tests is wholly inappropriate. Secondly, the morphological data appear to be from a single animal. If unsure, consult a statistician.

Minor points:

Be consistent in reporting of P values. Some are in the text, other the figure legends.

p4, "...therefore represent molecular innovations acquired after divergence of the fish lineage." this is only one possibility as gene loss in fish could be another.

Reviewer #4 (Remarks to the Author):

I have a negative opinion on this manuscript. **[Comment removed by Editorial Team as per Author Request]** It starts with the abstract which simply does not form a logical narrative. Perhaps, some of these results can be publishable but in a completely different form. There has to be one story which the manuscript tells and all results have to be arranged around this single narrative. Right now,

I simply cannot understand what the point of the manuscript is. Proteomics results should probably be published completely separately to electron microscopy. If the point of the manuscript is to evaluate the impact of whole genome duplication on synaptic proteome then there has to be a comparison to at least one (ideally more) invertebrate proteomes.

Reviewers' comments:

We thank the reviewers for their constructive comments and for recognizing the paper as “a valuable resource to the field”, “I would support publication of this paper based on its merit as a resource”, “a very interesting paper that would provide considerable information for future studies in synapse structure/function relationships” and that “the data also has the potential to broaden the fields understanding of a minimal functional unit of a vertebrate synapse and modifications of this unit in different lineages”. We are also grateful for their complements regarding the presentation; “the manuscript is clearly written” and the ultrastructural work is “beautiful and clearly presented”.

We have addressed all the points raised, generated new data and rewritten the manuscript according to the reviewer’s directions. Below we present the reviewer’s comments in italics followed by our point-by-point responses.

Reviewer #1 (Remarks to the Author):

Bayes et al present a comparative analysis of mouse and zebrafish synaptic proteomes. They find that the zebrafish PSD proteome is of smaller size than that found in mammals. The lower complexity of zebrafish PSDs results from fewer protein families. Interestingly, a major difference between mouse and zebrafish PSD proteomes is an enrichment in proteins involved in endocytosis, the regulation of vesicle transport, the actin cytoskeleton, and adherens junctions, in mouse PSDs. Together, these findings suggest that zebrafish synaptic proteomes have a reduced functional repertoire compared to the mammalian proteomes.

Overall, the data presented in this paper and made available in the genes2cognition database are a good reference database for zebrafish researchers and a valuable resource to the field. It is the first paper describing the zebrafish synaptic and PSD proteome, allowing a better understanding of vertebrate PSD complexity and evolution. The bioinformatics approach and methodology used seem sound. The manuscript is clearly written. The paper is essentially descriptive in nature: zebrafish synapses are studied at the ultrastructural and proteome level. Its main conclusion, that zebrafish PSDs have a reduced functional repertoire compared to their mammalian counterpart, is not functionally tested. However, this is not a trivial thing to do and I would support publication of this paper based on its merit as a resource.

Comments:

- 1) *The authors mention that proteomic profiling of zebrafish and mouse synaptic*

proteomes was performed in triplicate. Please clarify in the text whether all brains of mouse or zebrafish were pooled and split into 3 identical samples or whether the whole experiment was performed 3 times independently.

The whole experiment was performed 3 times independently and thus the three replicas are biological replicates. We have amended the materials and methods to make this clear.

Revised Methods text: "Three independent biological replicas were prepared for both zebrafish and mouse brain samples and each contained 0,7-1g of tissue".

2) A large difference was observed in the number of significantly enriched terms between mouse and zebrafish PSD proteomes: mouse has more GO terms than zebrafish. Is it possible that the mouse GO annotations database is more complete and comment.

We had in fact taken in consideration the difference in genome annotation by performing a double enrichment analysis with Zf-sPSD proteins. These proteins were first analyzed against the zebrafish genome and later their mouse orthologs were used for an enrichment analysis against the mouse genome. This data was included in the Supplementary Table 7 (Third sheet) but not clearly explained in the main text. This supplementary table has two lists of enriched terms. The first list, titled 'Annotated in Zebrafish Genome', contains enriched terms calculated using zebrafish protein against zebrafish genome. The second list, titled 'Annotated in Mouse Genome', contains enriched terms calculated using mouse orthologs of zebrafish PSD specific proteins against the mouse genome. Collapsing the terms enriched in these two analyses retrieves the number of enriched terms referred to in the manuscript (17 'Biological Process' terms and 8 'Cellular Component' terms). We believe that this double analysis importantly minimizes the bias caused by different genome annotation quality and strengthens the observation that Zf-sPSD are not highly related at the functional level.

We have added the following sentence into the corresponding results section to clarify this point:

"Furthermore, to account for the possible difference in annotation of the zebrafish and mouse genomes, the enrichment analysis with zebrafish proteins was done twice, first using Zf-sPSD proteins against the zebrafish genome and later using mouse orthologs of Zf-sPSD against the mouse genome. The final list of zebrafish enriched terms corresponded to the sum of terms enriched in either analysis."

3) *The main finding that mouse PSDs are enriched in SNARE, ESCRT and HOPS proteins compared to zebrafish PSDs is interesting and could be further explored. How do these findings compare to rat and human PSDs? Supp Table 7 lists GO enrichment analysis with human and rat proteins, but it is not easy to cross compare the findings with zebrafish. Can these results be shown in a table side by side with the zebrafish data? Independent validation of these findings, using for instance western blot, would enhance these findings.*

We have generated the table as requested and added it to the manuscript (Suppl. Table 8). We have found that human and rat PSDs present many more components of these complexes than zebrafish, with the exception of ESCRT complexes in rat. We have also looked into a very high quality and independently generated PSD proteome from mouse (Distler U et al. Proteomics. 2014 Nov;14(21-22):2607-13) and again found many components of these complexes (also included in Suppl. Table 8). Finally, the presence of ESCRT complexes at the PSD is supported by a recent publication (Chassefeyre R, J Neurosci. 2015 Feb 18;35(7):3155-73). Taken together all these findings suggest that these protein complexes reside at the mammalian PSD.

We have added the following sentence to the manuscript (Results section: Species specialization in the PSD) to account for these new results:

“To further explore this observation we looked for these proteins in the PSD from human, rat and an independent and recently generated mouse PSD proteome. In all species we did find more representatives of all these protein complexes than in zebrafish (Suppl. Table 8) with the exception of ESCRT components in rat.”

Mass spectrometry gives unequivocal protein identification and western blots will provide little additional information. Furthermore, performing western blots on the proteins from Figure 5 would represent a significant effort since validated specific zebrafish antibodies are a major and well-recognized problem for zebrafish research. For instance, we did try 7 antibodies for well characterized synaptic proteins present in our proteomics data (Psd95/Psd93, Psd95, Psd93, Sap102, Camk2a, Synaptophysin and Syngap1) to further validate some of our findings and only got one to work with zebrafish samples (Affinity ref. MA1-045), which recognizes Psd95 and Psd93 (Suppl. Fig. 4d). The reference database for zebrafish biologists (zfin.org) contains updated information on antibodies validated for use with zebrafish samples. Of the 75 proteins listed in the new Suppl. Table 8 only 7 have a validated antibody in this database. Additionally, of these, only 1 (Vamp2) would really be useful to show species-specific differences in PSD composition, as the rest (Stx1a, Stx4, Snap25, Chmp1a, Ist1 and Vps11) are either absent or present in both species PSD (see Fig. 5).

4) *Can the authors comment in the Discussion on the implications of their findings for the use of zebrafish as a model of human synaptopathies? Their findings suggest that a main difference between mammalian and zebrafish synaptic proteomes may be that the former have a much more complex repertoire of proteins involved in regulated protein trafficking. How does this impact the use of zebrafish as a model in human brain disorder, neuropharmacology and neurotoxicology research, as outlined in the opening paragraph of the Introduction?*

We welcome this comment and have added relevant text in the discussion. In the editing of the manuscript requested by another reviewer we have reduced the emphasis from disease in the opening paragraph and rewritten the introduction.

Reviewer #2 (Remarks to the Author):

This study aims to characterize the zebrafish synapse/PSD structures and their proteomes. Comparison to mouse PSD proteome suggests a lower complexity of zebrafish PSD with fewer protein families comprising of more gene duplicates.

The authors have performed extensive EM and quantitative proteomics experiments. The approaches and the subsequent analyses are well established. The conclusion and the references are appropriate. I am not aware of any major fault in this study. However, this study is purely descriptive. It does not provide useful data that advance our understanding of zebrafish behaviour or the use of zebrafish as a model for animal behavioural/brain disorder studies.

We thank the reviewer for their supportive comments.

We are concerned that the statement “However, this study is purely descriptive” is being labeled as a criticism. Firstly, we recognize the value of “descriptive” research, such as those reporting genome sequencing, of which there are many very important examples in *Nature* publications. Secondly, the paper is not purely descriptive as we have set out to test a hypothesis: that the TSGD results in increased synapse proteome complexity. As described in the paper, we confirm this hypothesis for the overall synapse proteome but show this hypothesis does not hold up for the postsynaptic density proteome.

We would disagree with the last statement “*It does not provide useful data that advance our understanding of zebrafish behaviour or the use of zebrafish as a model for animal behavioural/brain disorder studies*”. We are confident that this dataset will advance our understanding of zebrafish behavior and its use in disorders because we have established this in on our extensive experience with similar datasets in mice and

humans. For example, our human PSD proteome data has been used in four *Nature* Articles (amongst other papers) including human genetic papers of disease and intelligence (e.g. PMIDs: 24463508, 24463507, 21784246, 24399044), the analysis of the human brain transcriptome (PMID: 22996553) and single cell transcriptomes (26858593). Our work on mouse synapse proteomes has also been used with mouse genetic papers, including disease models and studies of vertebrate genome duplications in behavior and physiology (e.g. PMIDs: 23201971, 23201973). We have also shown how particular behaviours are enriched in the postsynaptic proteome using human and mouse phenotype ontologies (PMID: 21170055). We therefore fully expect that this resource will be valuable for zebrafish biologists and have amended the discussion to aid the reader by mentioning some of these examples. We also note that reviewer 3 said “Overall this is a very interesting paper that would provide considerable information for future studies in synapse structure/function relationships”.

There are several minor points that deserve discussion.

1) The authors highlight the lower complexity of zebrafish PSD proteome. I would define PSD proteins as proteins either unique or enriched in PSD (compared to synaptosome). As such, zebrafish PSD proteome contains only 10% less proteins. Is it still significantly differ from mouse PSD?

As suggested by the reviewer, we have computed if the zebrafish PSD proteome, defined only by proteins unique or enriched in it, would still be significantly smaller than that found in mouse. Using the same statistical approach we have again seen that in this smaller zebrafish PSD we identify less proteins than in the rodent species ($p = 0.0003$, binomial test).

We have added a sentence to the text incorporating this observation (Results section: Proteomic profiling of zebrafish and mouse synapses):

“This difference is still significant if the PSD is defined as the sum of zebrafish proteins exclusively found in the PSD or significantly enriched in it ($p = 0.0003$ binomial test).”

2) It should also be realized that the comparison of protein abundance levels may be hampered by the biochemical challenges in extracting the same synapse material from the vastly different zebrafish and mouse brains.

We acknowledge the relevance of this comment and were well aware of it during our experiments. We have several lines of evidence indicating that we had a similar biochemical extraction performance between species:

1. Protein yield, which measures the amount of protein recovered per unit of tissue after a biochemical fractionation, was not different between species. Neither for the synaptosomal fraction nor the postsynaptic density (Suppl. Fig. 4b,c).
2. Enrichment of the well-known postsynaptic density markers Psd95 and Psd93 in the PSD was also equivalent between species, as assessed by western blot (Suppl. Fig. 4d).
3. Proteins unique or enriched in the PSD contain very well known PSD components in both species. These include: glutamate receptors (AMPA, NMDA and Delta), protein adapters/scaffolds (including those from the families of: Baiap, Begain, Dlg, Dlgap, Homer, Lin7 or Magi), Small GTPases (including those from the families: Rab, Rac or Ral), Small GTPases modulators (including those from the following families: Arhgap, Arhgef, Arfgap, Syngap, RabGef, Iqsec or kalirin), Kinases (Camk2s, Cdk15, casein kinases, Dap Kinases), actin binding proteins or voltage-dependant calcium channels.
4. Overall we found a large overlap of proteins enriched in the PSD between mouse and zebrafish (56%).
5. The functional repertoire of the complete SYN and PSD proteomes is quite similar between species (Fig. 4).
6. We have investigated if particular types/groups of PSD proteins are more common in mouse or zebrafish. To do this we have first identified the main protein groups amongst PSD proteins using GO-Slim (Figure R1 below). Next we looked for the number of peptides identified for each PSD protein and calculated the number of peptides per protein in each Go-Slim group. This value can be used as the abundance of any given group in one species. Finally we have calculated the ratio of peptides per protein between mouse and zebrafish (Figure R1). The vast majority of the 140 GO-Slim categories have a ratio very close to 1, indicating that these protein groups are found in similar amounts between species. Those terms showing larger differences are poorly related to the synapse (marked with an asterisk in Figure R1, i.e. Unfolded Protein Binding, Protein Folding, nucleocytoplasmic transport, nuclease activity or lipid particle).
7. Finally, we have compared zebrafish PSD protein abundance to an independently generated RNA sequencing analysis of adult zebrafish whole brain. We have found that many proteins absent from the zebrafish PSD are expressed at very low levels, or even not expressed, at the RNA level, further strengthening our observations. We have added a new Supplementary Figure (#8) with the RNAseq data. We have also added a comment in the manuscript describing this.

Despite all these observations, indicating a general equivalent performance in both species of the biochemical methods used, we can't fully rule out that for a small subset

of synaptic proteins these methods might not perform equivalently. For this reason we only comment on groups of functionally related proteins, as they are less likely to appear differentially expressed due to technical reasons. For instance, related proteins can be those belonging to the same GO category (such as those in Table 2) or belonging to the same protein complex (as those in Figure 5). Finally, we would like to mention that to consider a protein as exclusive to the PSD of one species we have established that it has to be absent from the other species PSD but also from its SYN proteome.

Taking all this into consideration we have introduced the following modifications into the manuscript:

1. A clearer exposition along the lines described above in the results
2. Results section: 'Proteomic profiling of zebrafish and mouse synapses': "These data indicates that the performance of the biochemical methods used is equivalent between species, although differential effects on small numbers of proteins can not be fully ruled out."

Figure R1 (next page). Protein abundance in main PSD groups between mouse and zebrafish.

The ratio of peptides per protein found in mouse vs. zebrafish is shown for the 140 GO-Slim categories enriched in PSD proteins. Dashed red line denotes a ratio of 1, indicating an equivalent enrichment in each category between species. Stars denote terms with twice or more peptides found in mouse over zebrafish.

3) *Figure 5 shows the presence of classic presynaptic exocytosis proteins in PSD. I suppose Stx1-Snap25-Vamp2 were highly depleted in PSD? These proteins are generally not considered as PSD proteins, and should not be highlighted as PSD proteins.*

While we fully agree with the reviewer that the main synaptic function of these molecules is at the pre-synaptic side, in line with our data several reports also place them at the postsynaptic side:

1. SNAP-25, a Known Presynaptic Protein with Emerging Postsynaptic Functions. Antonucci F, Corradini I, Fossati G, Tomasoni R, Menna E, Matteoli M. *Front Synaptic Neurosci.* 2016 Mar 24;8:7. **[Review]**.
2. Identification of the SNARE complex mediating the exocytosis of NMDA receptors. Gu Y, Haganir RL. *Proc Natl Acad Sci U S A.* 2016 Oct 25;113(43):12280-12285.
3. SNARE Protein Syntaxin-1 Colocalizes Closely with NMDA Receptor Subunit NR2B in Postsynaptic Spines in the Hippocampus. Hussain S, Ringsevjen H, Egbenya DL, Skjervold TL, Davanger S. *Front Mol Neurosci.* 2016 Feb 5;9:10.
4. Postsynaptic VAMP/Synaptobrevin Facilitates Differential Vesicle Trafficking of GluA1 and GluA2 AMPA Receptor Subunits. Hussain S, Davanger S. *PLoS One.* 2015 Oct 21;10(10):e0140868.

Nevertheless, as the reviewer anticipates Stx1a/STx1b are found significantly depleted in the PSD relative to the synaptosomal fraction. Thus, these proteins are reported as absent from the PSD (appear with a blue square in Figure 5). Regarding Vamp2 and Snap25 our data is not as conclusive. Both proteins appear more abundant in the synaptosomal fraction, yet this abundance difference does not reach statistical significance. Accordingly, we had to include them in the group of proteins found with 'Equal Abundance' in SYN and PSD and thus to classify them as PSD components. We consider that we should not remove these proteins from the PSD list as they comply with the criteria that we have established.

We have added the following sentence to the manuscript (Results section: Species specialization in the PSD) to account for this reviewer comment:

"We have also found in the PSD proteins with very well established pre-synaptic functions, such as Snap25 or Vamp2. While these might be biochemical contaminants of the PSD preparation, several recent publications have given evidence for their participation in postsynaptic processes. Thus their localization in the PSD can not be

excluded.”

Reviewer #3 (Remarks to the Author):

The manuscript entitled, "The zebrafish synapse proteome: A resource for evolutionary and comparative analysis of vertebrate brain complexity," provides a detailed comparison of the proteome from two different synapse preparations and between zebrafish and mouse. The manuscript includes ultrastructural and morphological analyses of synapses from a limited number of brain regions.

The sample preparation, proteomics and statistical analysis follow well-established protocols established by this group. Overall this is a very interesting paper that would provide considerable information for future studies in synapse structure/function relationships. The data also has the potential to broaden the fields understanding of a minimal functional unit of a vertebrate synapse and modifications of this unit in different lineages.

The relative merit of the proteomics remains high with only minor clarifications needed:

The notion that the fish synaptosome preparation contains fewer proteins is reasonable based on the data and interesting considering the selective retention of synaptic proteins following the WGD. Differences may also represent species specific utilization. But are there other explanations to consider.

The fish synaptosomes contains *more* proteins than the mouse and the PSD contained fewer. We therefore assume that the reviewer was referring to the PSD preparation and will respond accordingly. The analysis of other fish species reveals that indeed synaptic protein families are larger. To test if the findings are specific to zebrafish, we looked at gene family size in a number of additional species. When the ratio of number of homologs per gene family are considered, a clear clustering of the teleosts including Zebrafish is seen. As this clustering excludes the Spotted Gar a teleost that did not undergo the teleost WGD we believe that the body of evidence agrees that the expansion seen is due to gene gain in teleosts most likely through WGD rather than gene loss in non-teleost vertebrates. We have added text to the manuscript describing this and added a new supplemental figure (#6). This expansion in teleosts species has been exemplified in the new manuscript version with a new panel in figure 3 where orthology ratios are shown for selected genes coding for key synaptic proteins.

Additional text added to manuscript below:

“This ratio is also seen in other teleosts but not in the Spotted Gar (*Lepisosteus oculatus*) a fish whose lineage diverged before the teleost specific WGD²² (Suppl. Fig. 6). Examples of the increased ratio of orthologs in key synaptic proteins among fish species but not in the Gar are shown in Figure 3e. This data supports the assumption that the higher ratio of gene family size is due to expansion of zebrafish genome by the TGD rather than by gene loss in mammalian genomes.”

First, the homogenization and centrifugation methods used for the preparations is a standard protocol developed for mammalian tissues. Is the method be equally effective on a fish brain?

We agree with reviewers 2 and 3 that a different performance of the biochemical methods on mouse and zebrafish tissue might have an important effect on our results. Nevertheless, our data indicates that large differences are unlikely to have occurred for the reasons exposed in the second point raised by the second reviewer. Please refer to the explanations therein.

The absence of a specific gene may be a technical issue not a biological difference. Does the author have the ability to access the quality of the tissue homogenates from the fish beyond the one immunoblot provided? This is especially important considering the considerable lack proteins associated with vesicle trafficking and endocytosis which one might expect to be highly conserved.

As mentioned above we have compared mouse and zebrafish SYN/PSD samples using several approaches, reaching the conclusion that important differences are unlikely. We have seen very few protein types showing differences in abundance between species and these are not obviously related to synaptic function (See Figure R1 above).

We agree with the reviewer that the absence of a specific protein may be due to technical reasons. It is for this reason that we only comment on groups of functionally related proteins, as they are less likely to appear differentially expressed due to technical reasons. For instance, related proteins can be those belonging to the same GO category (such as those in Table 2) or belonging to the same protein complex (as those in Figure 5). Finally, we would like to mention that to consider a protein as exclusive to the PSD of one species it had to absent from the other species PSD but also SYN proteomes.

Secondly, along this line, the presence or absence of specific genes in zebrafish, presented in Fig 5 seems incomplete. A quick search of ensembl shows annotated sequences for the majority of the syntaxins and vsp's, and the zebrafish information

server provides in situ hybridization data for many of these genes, yet the boxes are white indicating not in fish. Is this a misreading by this reviewer, please clarify.

In Figure 5 a protein with a white box in the mouse or zebrafish lane denotes that it was not identified in our proteomics experiment. Nevertheless, as the reviewer indicates, most genes coding for these proteins are correctly annotated in the zebrafish genome. We therefore conclude that these proteins do not belong to the zebrafish synaptic proteome, although they exist in zebrafish.

We have added a legend into the figure and have made the figure legend more clear to avoid possible confusion.

Once clarified, can the authors elaborate on the relative importance of neofunctionalization in mammals versus gene loss in fish. In their analysis did the authors consider tandem duplication versus the GWD as a source of additional genes.

Although the teleost specific WGD would have doubled all genes in the genome, the majority of orthologs between mouse and zebrafish currently exist as 1:1 pairs (Fig.3d), suggesting that the dominant evolutionary force following WGD was loss of duplicates from the fish genome. Where ortholog expansion is seen, it tends to be in the ratio of 2:1 rather than 1:2 (zebrafish:mouse; Fig. 3d), reflecting the duplication seen after WGD. Importantly, as we have shown, many genes coding for synaptic proteins are amongst those retained as duplicates following the teleost specific WGD. Furthermore, few mouse genes coding for synaptic proteins do not have an orthologue in the zebrafish genome (3223/3549 ~ 90% of synaptic proteins have an orthologue between mouse and zebrafish). This also holds true for proteins exclusive to the PSD (we have included this information in the new version of the manuscript: Suppl. Information and Suppl. Fig.7). Whilst it is true that these mouse specific genes may be due to multiple independent gene gain and neofunctionalization events in the mammalian lineage this would have to occur very rapidly in evolutionary terms between the divergence of the teleosts and before the radiation of mammals. Thus we believe that the most parsimonious argument that requires fewer independent evolutionary events is WGD and gene loss in the teleost lineage. Likewise, the gene gain by tandem duplication of multiple spatially unrelated genes on multiple chromosomes is less parsimonious than the proposed WGD.

Additional evidence for whole genome duplication is now given by comparison of ratios of orthologue counts within gene families between mouse and other species including additional fish, invertebrate and chordate species (see figure below). This analysis shows clear clustering of the fish species as distinct to other chordates or vertebrates

(added to the manuscript as Suppl. Fig. 6 also shown below). Interestingly, the Gar (*Lepisosteus oculatus*) a teleost whose lineage diverged before the TGD (PMID: 26950095) clusters with vertebrate species rather than with fish. This data indicate that all fish species present similar structure of homologous pairs with mouse giving evidence for their common origin at the TGD. The expansion in the teleosts is clearest when key gene families are compared this is now added to the manuscript as additional panel to Figure 3e. Additional text added to manuscript below.

“This ratio is also seen in other teleosts but not in the Spotted Gar (*Lepisosteus oculatus*) a fish whose lineage diverged before the teleost specific WGD²² (Suppl. Fig. 6). Examples of the increased ratio of orthologs in key synaptic proteins among fish species but not in the Gar are shown in Figure 3. This data supports the assumption that the higher ratio of gene family size is due to expansion of zebrafish genome by the TGD rather than by gene loss in mammalian genomes.”

Supplementary Figure 6. Principal components analysis (PCA) of the ratio of homologs.

Biplot of the first two components of a PCA model comparing ratios of homolog counts in gene families for multiple species including additional fish.

Whole genome data was obtained from Ensembl and the number of homologs between mouse and each species for gene families were determined. The resulting matrix of homolog ratios for all species were compared using PCA.

The invertebrate and chordate species shows clear clustering away from the vertebrate species. This analysis supports the assumption that gene family size is dominated by vertebrate lineage whole genome duplication events. The separation of the Spotted Gar (*Lepisosteus oculatus*) whose lineage diverged before the additional teleost specific WGD⁶ places it in the same quadrant as the mammalian species supporting the premise that the major event in genome evolution of the teleost fish was the additional WGD event.

Major concerns which need be addressed.

The ultrastructural data, beautiful and very clearly presented, is not tied into the major points of the manuscript. A link between the differences in distribution of the types of synapses observed and potential differences in proteomics profiles is eluded to and would have been interesting as a follow-up, but these cannot be made since the whole brain was used in the proteomics. Any variation in protein expression within or across species or association with a specific class of synapses are lost.

We thank the reviewer for acknowledging the EM data. In the redrafting of the introduction and results we have explained the importance of showing the ultrastructure alongside the proteomics. The history of the field is that the PSD was first observed using EM and a purification strategy then developed based on the tracking of the EM structure in biochemical fractions. Thus, it is essential to demonstrate that zebrafish have a PSD in brain synapses before applying this biochemical protocol. We believe the redrafting has improved the coherence of the paper considerably.

We completely agree that linking the morphologies with specific proteins would be ideal. It still remains a very difficult challenge at the ultrastructural level to make molecular correlates, however there are new approaches at the super-resolution level (e.g. PMID: 27109929) where synapse morphologies can be linked to postsynaptic protein labeling and distribution at the sub-synaptic levels. This is well beyond the scope of the current manuscript.

Although this may be be correct that these are the first EM analysis of these brain regions in an adult zebrafish, the authors failed to cite any relevant work from other species. A quick web search popped up numerous examples of ultrastructural analysis from the 1970's and 80's of a considerable many regions of the teleost brain included those in this manuscript, with descriptions of synapse types, locations on various cell

structures, morphometric analysis and variations across strata. Please cite appropriate published work.

We thank the reviewer as, indeed, we had overlooked the older literature and missed relevant references from other bony fish species from the 1970's and 80's. We have modified the manuscript accordingly and incorporated new references.

In the methods it seems a single animal was used for the morphological analysis which limits the ability to draw large conclusions from the sample. Better description and possibly images of the low magnification images indicating the precise region used for the ultrastructure is needed so as not to hinder a readers ability to replicate the findings.

Electron microscopy studies were performed with two animals to replicate initial observations. The methods section has been amended to make this point clearer.

In the supplementary information we had included an extended analysis of the EM data and a figure with 4 large panels (Suppl. Fig.1) one for each of the 4 brain regions studied: olfactory bulb, telencephalon, optic tectum (OT) and cerebellum (CC). In each of these panels a low magnification image of a coronal section of the entire brain region is included. This is stained in toluidine blue. In the olfactory bulb and telencephalon asymmetric synapses (with a PSD) were observed across the entire section. But in OT and CC asymmetric synapses were only observed in certain regions. In these two later regions we used the low magnification images to indicate where asymmetric synapses were found using EM.

Supplemental methods states, "Mean lengths and areas were compared by an analysis of variance (ANOVA) test and differences between groups were identified through the Tukey post hoc test. Significant differences between groups were checked with Kruskal-Wallis non-parametric test." There are several problems with this statement. First, an ANOVA followed by Tukey are parametric analysis that makes several assumption such as the data are normally distributed. KW in a non-parametric analysis. The authors must decide which type of test is allowed by the experimental design. Secondly analysis by two different tests is wholly inappropriate. Secondly, the morphological data appear to be from a single animal. If unsure, consult a statistician.

We have revised the statistics done with EM data and corrected them so that all analysis were performed with a non-parametric test (Kruskal-Wallis test). This has nevertheless not affected the results or conclusions of the EM data.

We have changed the methods section accordingly:

“Groups were compared through median lengths and areas and significant differences were analysed through the Kruskal-Wallis non-parametric test.”

Minor points:

Be consistent in reporting of P values. Some are in the text, other the figure legends.

We have remove text p-values and leave those in figures except in those cases in which there is no figure to support the data presented.

p4, "...therefore represent molecular innovations acquired after divergence of the fish lineage." this is only one possibility as gene loss in fish could be another.

Taking into account that 3223 out of the 3579 proteins found in mouse have an orthologue at the zebrafish genome, which accounts for 90% (Suppl. Fig. 7), gene loss would only be responsible for a small fraction of mouse specific proteins. Nevertheless, we have modified the text to take into consideration this possibility:

“These represent molecular innovations either acquired by mammals after divergence of the fish lineage or specifically lost from the fish lineage.”

We have also included an extra section in the Supplementary Information (C. Evolutionary origins of species differences in SYN and PSD proteomes) discussing this point, as refer to above.

Reviewer #4 (Remarks to the Author):

*I have a negative opinion on this manuscript. **[Comment removed by Editorial Team as per Author Request]** It starts with the abstract which simply does not form a logical narrative. Perhaps, some of these results can be publishable but in a completely different form. There has to be one story which the manuscript tells and all results have to be arranged around this single narrative. Right now, I simply cannot understand what the point of the manuscript is. Proteomics results should probably be published completely separately to electron microscopy. If the point of the manuscript is to evaluate the impact of whole genome duplication on synaptic proteome then there has to be a comparison to at least one (ideally more) invertebrate proteomes.*

Although this reviewer’s comments are at odds with the other three, we acknowledge

that the manuscript could be clearer and we have therefore substantially rewritten it. As recommended by the Editor, we have also emphasized the value of the data and website as a resource for the community in the rewritten version.

We also wish to explain to the reviewer that there are three major points to our research and findings:

The first point is that despite considerable research on mammalian synapse proteomes, the fish synapse proteome is uncharacterized. This is an important issue for many reasons. The study of the mouse postsynaptic proteome and comparison with *Drosophila*, which we published in 2008 (PMID: 18536710), revealed the “vertebrate expansion” in complexity. At the time of publication, the mechanism for this expansion was unknown and it later became clear that it was two rounds of genome duplication at the base of the vertebrate lineage. Subsequent experimental genetic experiments formally established that the generation of ohnologs and their diversification was a driver of behavioural and electrophysiological complexity (PMID: 23201971, 23201973). As teleosts experienced a third round of whole genome duplication, then it might be assumed that this would increase synaptic proteome further. This is the hypothesis that we have tested in this manuscript. We found that the overall synapse proteome increased in zebrafish proportionally to the increase in gene number as a result of the teleost specific WGD. Unexpectedly, we observed a reduction in the postsynaptic density proteome.

The second point is that it is essential to understand the synapse proteome complexity of fish if they are to be used as model organism for human diseases. The greater number of fish paralogs has many implications. For example, multiprotein complexes such as neurotransmitter receptors and multiprotein signaling pathways are assembled by subunits and multiplicative combinatorial expansion may occur in teleosts. The combinatorial possibilities and subunit redundancy will all affect phenotypes of mutations. From studies of human synapse proteomes we now know that there are over 400 synapse genes known to carry disease variants involving over 130 diseases. It is not straightforward to simply match the ortholog in fish and humans and assume this will be valid disease model when there are so many other molecular differences.

The third point is to justify why we have included the EM study. The history of the field is that the PSD was first observed using EM and a purification strategy was then developed based on the tracking of the EM structure in biochemical fractions. Thus, it is essential to demonstrate that zebrafish synapses present PSDs before applying this biochemical protocol. In this paper we provide the first evidence for the existence of PSDs in the zebrafish brain. Moreover, rather than simply showing a single example of

a PSD in a fish sample, we more quantitatively examined them across multiple brain regions and identified features that were similar between mammals and fish. This is also important in our narrative structure because the proteomic studies identify a conserved “vertebrate PSD proteome” that is shared between species.

Regarding the comparison with an invertebrate proteome. We have previously compared the postsynaptic proteomic repertoire of fruitfly (*D. melanogaster*) with that of mouse (PMID: 18536710), which highlighted the trend for increased complexity of the mammalian synaptosome machinery. This current manuscript focuses for the first time on a defined non-mammalian PSD and synaptosome. The legacy of the well evidenced multiple rounds of WGD on the complexity of the PSD/synaptosome was investigated. A comprehensive analysis of all proteomes would require isolation of these from multiple organisms. However, the computational analysis of ortholog counts within gene families provides a proxy for gene gain/loss in different groups. Comparison of multiple invertebrate/vertebrate/chordate species (see Supplementary Figure 6 above, which has been incorporated into Supplementary information) highlights the distinction of all examined fish species which underwent the teleost specific WGD (Zebrafish, Tetraodon, Stickleback and Cod) with a bony-fish species which did not (the Spotted Gar, PMID: 26950095), invertebrates (fly, worm), chordates (Ciona species) and mammalian vertebrates (Human, Rat). Thus, supporting the premise of the whole genome duplication as a dominant factor in the evolutionary history of the teleost fish including zebrafish. We have added text to the manuscript describing this and added a new supplemental figure (S6). This expansion in the teleosts is clearest when key gene families are compared this is now added to the manuscript as additional panel to Figure 3. Additional text added to manuscript on this particular point is shown below.

“This ratio is also seen in other teleosts but not in the Spotted Gar (*Lepisosteus oculatus*) a fish whose lineage diverged before the teleost specific WGD²² (Suppl. Fig. 6). Examples of the increased ratio of orthologs in key synaptic proteins among fish species but not in the Gar are shown in Figure 3e. This data supports the assumption that the higher ratio of gene family size is due to expansion of zebrafish genome by the TGD rather than by gene loss in mammalian genomes.”

We trust the reviewer will find the revised version explains these points more clearly than before.

REVIEWERS' COMMENTS:

Reviewer #1 (Remarks to the Author):

The authors have made a serious effort to clarify my previous concerns. Their rewriting efforts have also substantially improved the manuscript. I have no further concerns and, as noted in my review of the initial submission, support publication on its merit as a valuable resource to the field.

Reviewer #2 (Remarks to the Author):

The manuscript can be accepted for publication.

Reviewer #3 (Remarks to the Author):

The authors' response is very thorough. The remaining two issues: The data of comparisons between mouse and zebrafish reported in Fig 5 remain a central tenant of the results but most if not nearly all proteins are not independently verified. The citations for the EM have improved but the data remain disjointed from the major points of the paper and hard to link to the proteomic analysis. Secondly these data remain buried in supplemental data.

Reviewer #4 (Remarks to the Author):

I find the presentation of the manuscript somewhat improved. The reply letter is adequate. The aims of the study are now closer to be clear. The discussion of the impact of whole genome duplications on the complexity of synapse proteomes is satisfactory. The authors have clearly made a lot of effort to reply to reviewer's comments.

However, I am still sceptical if these findings are sufficiently novel, well presented, technically sound, and of general interest to be publishable in a high IF journal like Nature Communications. On the basis of the merits of the bioinformatic analysis of gene duplications and losses alone, I would argue that this work should be submitted to a much more specialized journal (perhaps a proteomics journal). There is little new here in terms of gene or genome duplication.

[Comment removed by Editorial Team as per Author Request]

I am also worried that the findings on decreased complexity of the post-synaptic density (PSD) go against the main message of the study: that fish genome duplication resulted in more complex synapses. This is tricky: we are mixing two levels of description here, one from structural genomics (genome duplication), and the other from functional genomics and proteomics. Are these genes in the zebrafish genome, yes or no? If they are there, do they get expressed in neurons, do they make it to the PSD? The authors do not even seem certain if they were not missed somehow due to the way PSD were isolated biochemically. As there are so many complexities here, the writing and presentation would have to be very clear, and I fear they are not.

MINOR COMMENTS:

[Comment removed by Editorial Team as per Author Request] How was the tree in Fig 1a drawn? How was the tree rooted? The use of arrows on different branches does not make sense. Please, draw a real phylogenetic tree! Instead of "2 WGD" it should be "2R-WGD".

Response to reviewer's comments.

Reviewer's comment shown in italics.

Reviewer #1 (Remarks to the Author):

The authors have made a serious effort to clarify my previous concerns. Their rewriting efforts have also substantially improved the manuscript. I have no further concerns and, as noted in my review of the initial submission, support publication on its merit as a valuable resource to the field.

We thank the reviewer for his/her support to our work and for acknowledging our effort to respond to all reviewer comments.

Reviewer #2 (Remarks to the Author):

The manuscript can be accepted for publication.

We thank the reviewer for his/her support to our work.

Reviewer #3 (Remarks to the Author):

The authors' response is very thorough. The remaining two issues: The data of comparisons between mouse and zebrafish reported in Fig 5 remain a central tenant of the results but most if not nearly all proteins are not independently verified. The citations for the EM have improved but the data remain disjointed from the major points of the paper and hard to link to the proteomic analysis. Secondly these data remain buried in supplemental data.

We thank the reviewer for acknowledging our effort to respond to all reviewer comments and we have now addressed the two remaining issues:

To answer the first issue, we wish to make two points. First, we think the reviewer is overemphasizing the importance of Fig 5 because the key finding on PSD differences between zebrafish and mammalian species is obtained from the GO and KEGG enrichment analysis (shown in Suppl. Table 7 and in Table 2) - the data in Figure 5 is used to highlight some key proteins and protein complexes found from these enriched terms. Secondly, we have in fact already addressed the issue of independent verification in the submitted manuscript with RNAseq data showing the abundance of transcripts encoding the zebrafish synaptic proteins (Supplementary Figure 11). This RNA expression data was independently generated from the proteomics study. We fear that this new information might not have been clearly incorporated into the main text or might have not been visible enough and we have rectified this by editing the manuscript with the following changes:

- Retained Suppl. Fig. 11.
- Modified Figure 5 so as to show the RNA abundance data alongside the protein abundance data.
- Rewritten text in the Results and Discussion (See below)

We also wish to point out to the Editor that the editing of the results and discussion also satisfies the points raised by reviewer 4.

We have added the following text to the results:

“To further investigate the depletion of some mammalian proteins from the zebrafish PSD, we asked if the orthologous genes encoding these proteins were present in the zebrafish genome and if so, whether they were expressed at low levels. Of the 745 mouse PSD proteins absent from zebrafish synaptic proteomes (SYN+PSD), 80% have orthologs in the zebrafish genome (Supplementary Fig. 10). To examine the possibility that those might be expressed at low levels in zebrafish we examined the 84 proteins shown in Figure 5 using RNAseq data and found the expression of most of these genes ($63/84 = 75\%$) is low (less than 10 transcripts per million (TPM)) or very low (less than 1 TPM) (Fig. 5 and Supplementary Figure 11). The percentage of lowly expressed genes in this protein-depleted group is greater than we see for all SYN and PSD encoding genes, where the percentage of detectable genes with $TPM < 10$ is 51% and 53% respectively. This suggests that proteins depleted from the SYN and PSD of zebrafish show a corresponding low expression of encoding mRNA. Together these findings indicate that both low levels of expression and absence of orthologs are mechanisms contributing to the depletion of synapse proteins from the zebrafish synapse.”

We have also added the following text to the discussion:

“Our observations that there were fewer orthologs and low levels of mRNA expression suggest that gene loss and transcriptional regulatory changes are contributing mechanisms. In addition, there may be post-translational mechanisms such as protein stability. Although we cannot fully exclude the contribution of technical reasons, the observed enrichment in particular synaptic functions in the set of mammalian-specific PSD proteins suggests that rather than a technical issue, there are biologically relevant differences between zebrafish and mouse PSD.”

Secondly, the issue that most of the EM data is in supplementary files.

In the supplementary information we show a total of 36 EM images and have placed 6 images we thought were of most interest into the main Figure 1. We would seek the Editor’s guidance as to whether we should create an additional main figure that includes some of the material in supplementary.

Reviewer #4 (Remarks to the Author):

I find the presentation of the manuscript somewhat improved. The reply letter is adequate. The aims of the study are now closer to be clear. The discussion of the impact of whole genome duplications on the complexity of synapse proteomes is satisfactory. The authors have clearly made a lot of effort to reply to reviewer's comments.

We are glad that we had addressed reviewer's concerns.

However, I am still sceptical if these findings are sufficiently novel, well presented, technically sound, and of general interest to be publishable in a high IF journal like Nature Communications. On the basis of the merits of the bioinformatic analysis of gene duplications and losses alone, I would argue that this work should be submitted to a much more specialized journal (perhaps a proteomics journal). There is little new here in terms of gene or genome duplication.

[Comment removed by Editorial Team as per Author Request] *I am also worried that the findings on decreased complexity of the post-synaptic density (PSD) go against the main message of the study: that fish genome duplication resulted in more complex synapses. This is tricky: we are mixing two levels of description here, one from structural genomics (genome duplication), and the other from functional genomics and proteomics. Are these genes in the zebrafish genome, yes or no? If they are there, do they get expressed in neurons, do they make it to the PSD? The authors do not even seem certain if they were not missed somehow due to the way PSD were isolated biochemically. As there are so many complexities here, the writing and presentation would have to be very clear, and I fear they are not.*

We agree with the reviewer that we could improve clarity and have added the following text to the results that answer the reviewer's questions:

"To further investigate the depletion of some mammalian proteins from the zebrafish PSD, we asked if the orthologous genes encoding these proteins were present in the zebrafish genome and if so, whether they were expressed at low levels. Of the 745 mouse PSD proteins absent from zebrafish synaptic proteomes (SYN+PSD), 80% have orthologs in the zebrafish genome (Supplementry Fig. 10). To examine the possibility that those might be expressed at low levels in zebrafish we examined the 84 proteins shown in Figure 5 using RNAseq data and found the expression of most of these genes (63/84 = 75%) is low (less than 10 transcripts per million (TPM)) or very low (less than 1 TPM) (Fig. 5 and Supplementary Figure 11). The percentage of

lowly expressed genes in this protein-depleted group is greater than we see for all SYN and PSD encoding genes, where the percentage of detectable genes with TPM < 10 is 51% and 53% respectively. This suggests that proteins depleted from the SYN and PSD of zebrafish show a corresponding low expression of encoding mRNA. Together these findings indicate that both low levels of expression and absence of orthologs are mechanisms contributing to the depletion of synapse proteins from the zebrafish synapse.”

We also point out in the discussion that additional mechanisms as well as technical issues might also contribute. We think this gives a clear and thorough overview of the issues raised by the reviewer.

“Our observations that there were fewer orthologs and low levels of mRNA expression suggest that gene loss and transcriptional regulatory changes are contributing mechanisms. In addition, there may be post-translational mechanisms such as protein stability. Although we cannot fully exclude the contribution of technical reasons, the observed enrichment in particular synaptic functions in the set of mammalian-specific PSD proteins suggests that rather than a technical issue, there are biologically relevant differences between zebrafish and mouse PSD.”

MINOR COMMENTS:

[Comment removed by Editorial Team as per Author Request] *How was the tree in Figure 1a drawn? How was the tree rooted? The use of arrows on different branches does not make sense. Please, draw a real phylogenetic tree! Instead of "2 WGD" it should be "2R-WGD".*

We are grateful to the reviewer for raising this as there was an error in our diagram. The arrows at the end of the tree branches were inadvertently included - these have been removed. We have also substituted "2 WGD" with "2R-WGD", as suggested by the reviewer.

We wish to emphasize that the tree is not meant to be a proper phylogenetic tree but just a scheme to help readers who are not familiar with the theory of whole genome duplication events in the evolution of vertebrate genomes. We consider that computing a proper tree for this scheme is absolutely not required. Similar schemes can be found in many scientific articles. For instance:

- Rapid genome reshaping by multiple-gene loss after whole-genome duplication in teleost fish suggested by mathematical modeling. Inoue J, Sato Y, Sinclair R, Tsukamoto K, Nishida M. Proc Natl Acad Sci U S A. 2015 Dec 1;112(48):14918-23.
- Identification of Ohnolog Genes Originating from Whole Genome Duplication in Early Vertebrates, Based on Synteny Comparison across Multiple Genomes. Singh PP, Arora J, Isambert H. PLoS Comput Biol. 2015 Jul 16;11(7):e1004394.
- Convergent gene loss following gene and genome duplications creates single-copy families in flowering plants. De Smet R1, Adams KL, Vandepoele K, Van Montagu MC, Maere S, Van de Peer Y. Proc Natl Acad Sci U S A. 2013 Feb 19;110(8):2898-903.